# When Foundation Models are One-Liners: Limitations and Future Directions for Time Series Anomaly Detection

**Xiaokun Zhu, Louis Carpentier**[*] **& Mathias Verbeke**
KU Leuven, Department of Computer Science, Belgium
Leuven.AI - KU Leuven Institute for AI, Belgium
Flanders Make@KU Leuven
`{xiaokun.zhu,louis.carpentier,mathias.verbeke}@kuleuven.be`

## Abstract

Recent efforts have extended the foundation model paradigm from natural language to time series, raising expectations that pre-trained time-series foundation models generalize well across downstream tasks. In this work, we focus on time-series anomaly detection, in which time-series foundation models detect anomalies based on the reconstruction or forecasting error. Specifically, we critically examine the performance of five popular families of time-series foundation models: MOMENT, Chronos, TimesFM, Time-MoE, and TSPulse. We find that for each model family using varying model sizes and context window lengths, anomaly detection performance does not significantly differ to simple one-liner baselines: moving-window variance and squared-difference. These findings suggest that the key assumptions underlying reconstruction-based and forecasting-based methodologies for time-series anomaly detection are not satisfied for time-series foundation models: anomalies are not consistently harder to reconstruct or forecast. The results suggest that current approaches for leveraging foundation models in anomaly detection are insufficient. Building upon our insights, we propose alternative directions to effectively detect anomalies using foundation models, thereby unlocking their full potential for time-series anomaly detection.

## 1 Introduction

A *time series* is a sequence of data points that quantify any process evolving over time. It is a ubiquitous concept, since time is an intrinsic dimension of the world in which we all live. Many practical tasks revolve around time series, e.g., forecasting (Lim & Zohren, 2021), classification (Middlehurst et al., 2024), and anomaly detection (Schmidl et al., 2022). Time series have much in common with natural language because both spoken and written language are sequences of words and sentences. Moreover, tasks involving natural language often parallel those in time series analysis: question answering and dialogue systems (Gao et al., 2019) require predicting the next words for a given prompt (forecasting); spam detection and sentiment analysis (Minaee et al., 2021) involve assigning text to predefined categories (classification); and grammatical error detection (Bryant et al., 2023) aims to identify tokens that deviate from expected linguistic patterns (anomaly detection).

In recent years, a new paradigm has gained popularity in natural language processing: pre-training huge language models – often called foundation models or large language models (LLMs) – on massive text corpora on a single task, typically next-token prediction or sequence reconstruction (Zhao et al., 2025). After pre-training, the models do not only excel at the training task, but also demonstrate emerging abilities on other downstream tasks (Wei et al., 2022), even outperforming traditional methods designed specifically for those tasks (Devlin et al., 2019; Brown et al., 2020).

Given the commonalities between time series and natural language, researchers have recently begun applying the foundation model methodology to time series (Ansari et al., 2024a; Woo et al., 2024; Goswami et al., 2024; Shi et al., 2024). Mirroring the approach in natural language processing,

---

[*]Corresponding author

this involves pre-training time-series foundation models (TSFMs) on large-scale time series datasets using a single objective – typically forecasting or reconstruction – with the expectation that the resulting models can be adapted to a variety of downstream tasks and surpass task-specific methods.

One of those tasks is *time-series anomaly detection* (TSAD), the automatic detection of anomalous events in time series, which plays a critical role in many high-stakes applications such as fraud detection (Ferdousi & Maeda, 2006), fault diagnosis (Yan et al., 2024), and health monitoring (Pereira & Silveira, 2019), where timely identification of rare but significant deviations can prevent costly failures or save lives. Researchers have hypothesized the effectiveness of TSFMs in TSAD (Ansari et al., 2024a). Although preliminary experiments seem to confirm this (Goswami et al., 2024), thorough evaluations of these models are still lacking. A recent benchmarking study includes a more extensive evaluation of TSFMs and claims that TSFMs "show great promise", demonstrating "superior zero-shot capabilities when compared to most existing statistical and neural network-based methods" (Liu & Paparrizos, 2024). However, we argue that this study is insufficient for answering the following question: "are current approaches to apply TSFMs to TSAD effective?" On the one hand, the study heavily relies on aggregate metrics which overlook counterintuitive details in the results that lead to surprising findings, as we will demonstrate. On the other hand, Liu & Paparrizos (2024) analyses for each family of TSFMs with only a single model size and short context window lengths. As such, the study does not explore the sensitivity of the model size and context length, while these are two crucial factors for LLMs (Kaplan et al., 2020; Liu et al., 2023).

To address these limitations, we perform a thorough validation of four popular families of TSFMs on the task of TSAD: MOMENT (Goswami et al., 2024), Chronos (Ansari et al., 2024a), TimesFM (Das et al., 2024), and Time-MoE (Shi et al., 2024). We employ two mainstream methodologies: forecasting-based (Chronos, TimesFM and Time-MoE) and reconstruction-based (MOMENT), which derive anomaly scores from the model's forecasting or reconstruction errors (Schmidl et al., 2022). Additionally, we examine TSPulse (Ekambaram et al., 2025), which employs both methodologies and further incorporates frequency-domain inputs. For each model family except TSPulse, we include two variants of different sizes. Each variant is evaluated using at least three different context window lengths (64, 256 and 512). Our main contributions are summarized as follows:

- We show that the key assumption of reconstruction-based and forecasting-based anomaly detection – anomalies are harder to reconstruct/forecast – does not hold for TSFMs: under such error-based methodologies, their performance for anomaly detection is either similar to or not significantly better than one-liner baselines: moving-window variance (reconstruction-based TSFMs) and squared-difference (forecasting-based TSFMs). These results are irrespective of the model family, the model size, the context window length, or the model's reconstruction/forecasting performance.

- We suggest alternative directions to unlock TSFMs' potential for TSAD, such as exploiting long-term and short-term forecasting horizons, detecting anomalies in hidden representations, and performing end-to-end fine-tuning using labeled anomalies.

- We highlight the significance of visualizing both data and algorithm outputs for individual time series, which enabled our conclusions. This point has been previously articulated with eloquence (Wu & Keogh, 2023), yet it remains underappreciated within the community.

## 2    PRELIMINARIES AND RESEARCH QUESTION

A univariate time-series $T = (x_1, \ldots, x_n)$ is an ordered sequence of $n \in \mathbb{N}$ values $x_i \in \mathbb{R}$.[1] A subsequence $T_{s,e} = (x_s, x_{s+1}, \ldots, x_e)$ of $T$ with $1 \le s < e \le n$ of length $e - s + 1$ is a contiguous segment of $T$ starting at index $s$ and ending at index $e$. While many time-series analysis tasks exist, in this work we focus on time-series anomaly detection.

---

[1] We focus on univariate time series because the majority of TSFMs only offer native support for univariate inputs (Ansari et al., 2024a; Goswami et al., 2024; Shi et al., 2024; Das et al., 2024); thus, univariate evaluation is sufficient to assess their anomaly detection capabilities. Only recent exceptions, such as Chronos-2 (Ansari et al., 2025) and Toto 1.0 (Cohen et al., 2025), capture inter-channel dependencies. Preliminary qualitative results for those two models also align with our main findings and are detailed in Appendix M.

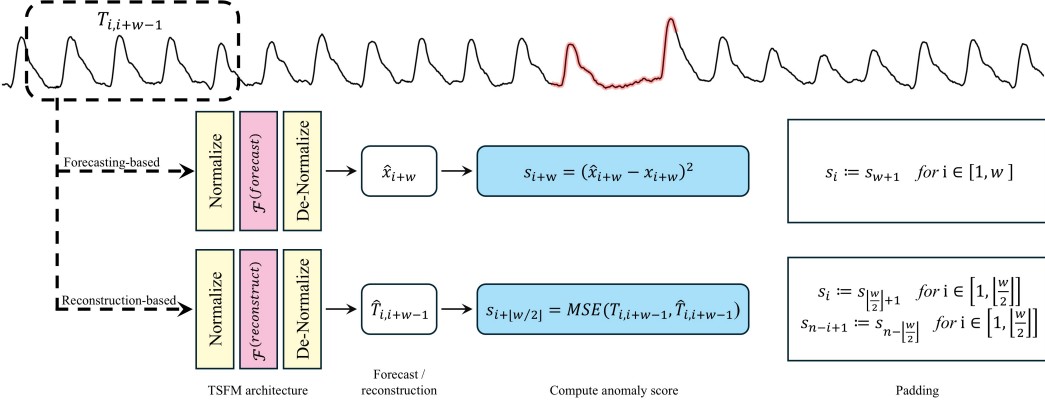

Figure 1: Application of TSFMs on TSAD. Given time series subsequence $T_{i,i+w-1}$, a forecasting-based TSFM predicts the next observation $\hat{x}_{i+w}$ and computes the anomaly score as the squared error with $x_{i+w}$. A reconstruction-based TSFM produces reconstruction $\hat{T}_{i,i+w-1}$ of the complete subsequence $T_{i,i+w-1}$ and computes the anomaly score as the mean squared error across all observations in the subsequence.

**Definition 1 (Time-series Anomaly Detection (TSAD))** *Given a time-series $T$ of length $n \in \mathbb{N}$, time-series anomaly detection is the task of predicting anomaly scores $s_i \in \mathbb{R}^+$ for each $x_i \in T$. For two observations $x_i, x_j \in T$, we say that $x_i$ is more anomalous than $x_j$ if $s_i > s_j$.*

Recently, numerous TSFMs have been proposed and some of them are reported to achieve state-of-the-art performance in TSAD (Goswami et al., 2024). While TSFMs share many similarities, a key design choice is the pre-training task. Most TSFMs are trained for forecasting, while others focus on reconstructing time series subsequences. To study the impact of this design choice on TSAD, we categorize TSFMs into forecasting-based and reconstruction-based models.

**Definition 2 (Forecasting-based Time-Series Foundation Model)** *A forecasting-based TSFM $\mathcal{F}^{(forecast)} : \mathbb{R}^w \mapsto \mathbb{R}$ is a deep neural network pre-trained to predict the next value following a context window of length $w$ using a massive set of time series.*

**Definition 3 (Reconstruction-based Time-Series Foundation Model)** *A reconstruction-based TSFM $\mathcal{F}^{(reconstruct)} : \mathbb{R}^w \mapsto \mathbb{R}^w$ a deep neural network pre-trained to reconstruct time series sequences of length $w$ using a massive set of time series.*

Forecasting-based TSFMs are commonly applied to TSAD under the assumption that anomalous observations are inherently more difficult to predict. Given a context window, the TSFM forecasts the next observation. If the predicted value deviates substantially from the actual observation, the point is considered anomalous. As a consequence, the forecasting error serves as an anomaly score. In contrast, reconstruction-based TSFMs assume that anomalous sequences are harder to reconstruct. They therefore detects anomalies by identifying the subsequences that deviate notably from their reconstruction, with the reconstruction error providing the anomaly score. Figure 1 illustrates these two approaches in greater detail. In this work, we analyze the effectiveness of these approaches for TSAD and answer the following research question:

**Research question** *Are current approaches to apply time-series foundation models to time-series anomaly detection effective?*

## 3 BACKGROUND

### 3.1 DESIGN CHOICES FOR TSFMS

Due to the similarities between time series and natural languages, many ideas of LLMs have been borrowed to construct TSFMs. Nevertheless, several adaptations to the architectures are necessary

to handle the time series characteristics. One of the main design choices is the learning tasks, as highlighted in Definitions 2 and 3, but there are also multiple other important choices such as the normalization, tokenization, embedding and architecture. Table 1 lists a summary of the made choices for each model family included in this study, and more details can be found in Appendix B. In particular, unlike the other four model families, TSPulse is a tiny (1M) non-Transformer model incorporating multiple heads: a reconstruction head in the time domain ($\text{Head}_{\text{time}}$), a reconstruction head in the frequency domain ($\text{Head}_{\text{fft}}$), and a forecasting head in the time domain ($\text{Head}_{\text{future}}$). For TSAD, these heads are combined via ensembling and triangulation, where the latter is a per-dataset model selection procedure based on the heads' performance on a tuning set (Ekambaram et al., 2025).

Table 1: Technical details of TSFMs

|  | **Chronos-Bolt** | **Time-MoE** | **TimesFM** | **MOMENT** | **TSPulse** |
|---|---|---|---|---|---|
| **Normalization** | Z-scaling | Z-scaling | Z-scaling | Z-scaling | RevIN |
| **Tokenization** | Patching | Point value | Patching | Patching | Patching |
| **Embedding** | Residual block | SwiGLU | Residual block | Linear projection | Linear projection |
| **Architecture** | Encoder-decoder | Decoder-only | Decoder-only | Encoder-only | TSMixer |
| **Pretrained on** | Forecasting | Forecasting | Forecasting | Reconstruction | Forec. + Recon. |

## 3.2 RELATED WORK

To our knowledge, the only extensive evaluation of TSFMs for TSAD is a benchmarking study published by Liu & Paparrizos (2024). They evaluated 32 univariate TSAD methods on the TSB-AD-U benchmark, which is introduced in the same paper to address common flaws identified in existing TSAD datasets (Wu & Keogh, 2023). Among the evaluated methods were five TSFMs: OFA (Zhou et al., 2023), Lag-Llama (Rasul et al., 2024), Chronos (Ansari et al., 2024a), TimesFM (Das et al., 2024), and MOMENT (Goswami et al., 2024). The evaluation conditions for these models varied: OFA was not pretrained on time-series data and requires explicit fine-tuning; MOMENT was evaluated in both zero-shot and fine-tuned settings; and the other three TSFMs were assessed only in a zero-shot setting. For fine-tuning, an initial segment of each time series was used as training data, which consists of only normal data, to improve either the reconstruction or forecasting performance of the TSFM. According to the results, MOMENT (Goswami et al., 2024) ranks among the top 5 for univariate TSAD, with a negligible performance difference between its fine-tuned and zero-shot evaluations. The three forecasting-based TSFMs (Lag-Llama, Chronos and TimesFM) were found to "excel at detecting point-based anomalies". Based on these results, the authors conclude that "the performance of time-series foundation models shows great promise" (Liu & Paparrizos, 2024).

## 4 EXPERIMENTAL SETUP

Our experiments cover five families of TSFMs: MOMENT (Goswami et al., 2024), Chronos (Ansari et al., 2024a;b), TimesFM (Das et al., 2024), Time-MoE (Shi et al., 2024), and TSPulse (Ekambaram et al., 2025). This selection includes both forecasting-based and reconstruction-based models and spans diverse model architectures. For each model family except TSPulse, we evaluate two instances of different sizes and use at least three different context window lengths to analyze the sensitivity of these parameters to the TSFMs' performance for TSAD. For TSPulse, we directly compare its results reported in Ekambaram et al. (2025) with our baselines under comparable conditions. Table 2 lists the details. We denote the experiment settings using the notation 'model family-model size-context window length', such as 'MOMENT-base-64'. We compare the TSFMs against two simple baselines to evaluate their effectiveness for TSAD. For reconstruction-based models, we use **moving-window variance**, as we found that model performance correlates positively with variance in anomalous regions (see Section 5.1). For forecasting-based models, inspired by the one-liners of Wu & Keogh (2023), we use **squared-difference**, which naively uses the mean of the neighboring observations as forecast, instead of a complex model.

**Definition 4 (Moving-window variance)** *Given a time series $T$ and subsequence length $w$, baseline Var-$w$ computes anomaly score $s_{i+\lfloor w/2 \rfloor} = \text{variance}(T_{i,i+w-1})$.*

Table 2: Experiment coverage

| Model | Liu & Paparrizos (2024) | | Ours | |
|---|---|---|---|---|
| | Size | Window | Size | Window |
| **MOMENT** | base | 64 | base, large | 64, 256, 512$^\dagger$ |
| **Chronos** | t5-base | 100 | bolt-small, bolt-base | 64, 256, 512 |
| **TimesFM** | 1.0 (200M) | 96 | 1.0 (200M), 2.0 (500M) | 64, 256, 512 |
| **Time-MoE** | not covered | | base, large | 64, 256, 512 |
| **TSPulse** | not covered | | 1M | 512 |

$^\dagger$For MOMENT-large, the window length 1024 was also covered.

**Definition 5 (Squared-difference)** *Given a time series $T$ and subsequence length $w$, baseline Last-$w$ computes anomaly score $s_{i+w} = (\text{mean}(T_{i,i+w-1}) - x_{i+w})^2$. Baseline Centered-$w$ computes anomaly score $s_{i+\lfloor w/2 \rfloor} = (\text{mean}(T_{i,i+w-1} \setminus x_{i+\lfloor w/2 \rfloor}) - x_{i+\lfloor w/2 \rfloor})^2$.*

For the datasets, we followed Liu & Paparrizos (2024) by using their TSB-AD benchmark because it is the latest curated benchmark emphasizing labeling quality. More specifically, we used the evaluation set of TSB-AD-U, which contains 350 univariate time series curated from 23 univariate datasets[2] (Liu & Paparrizos, 2024, Section B.1.1), to evaluate the models' zero-shot anomaly detection capabilities for univariate TSAD. We use the ID instead of the full name to denote individual time series in TSB-AD-U. For example, 'TSB-AD-U 811' denotes the time series with full name '811_Exathlon_id_2_Facility_tr_10766_1st_12590'.

For the metric, we followed Liu & Paparrizos (2024) by using VUS-PR because we believe it provides sufficient information for drawing our conclusions. As will become clear in our experimental results, while an evaluation metric can often serve as a shortcut for uncovering insights, all of our key findings can be derived by examining intermediate results, such as anomaly scores, or by conducting representative case studies, without relying on any specific metric.

## 5 EXPERIMENTAL RESULTS

We discuss our findings for reconstruction-based models (MOMENT) in Section 5.1, for forecasting-based models (Chronos, TimesFM, Time-MoE) in Section 5.2, for TSPulse in Section 5.3, and for non-TSFM anomaly detectors in Section 5.4. We only show representative experimental results without loss of generality; see Appendix E, F, K for complete results.

### 5.1 RECONSTRUCTION-BASED MODELS

**MOMENT-base is almost equivalent to moving-window variance.** Liu & Paparrizos (2024) shows that MOMENT-base using a window size of 64 achieved a remarkably high VUS-PR of 0.81 on the Exathlon dataset, making it rank among the top-performing anomaly detectors for this dataset. However, the mean anomaly length in Exathlon is 1577.3, which is almost 25 times larger than the context window length. We found that MOMENT-base performed well on time series in which the anomalous regions have higher variance than the normal regions when visualizing the Exathlon time series along with the predicted anomaly scores (Figure 2a). In contrast, MOMENT fails to detect anomalies which have a small variance (Figure 9a). This led us to suspect that MOMENT-base is in effect detecting the variance in its context window. To verify this speculation, we compare MOMENT-base with the baseline moving-window variance (Definition 4). Our speculation was consolidated by individual case study (Figure 2 and 9), one-to-one comparison (Figure 3), and statistical analysis (Table 3 and Appendix J). The root cause of this similarity is presented next.

**Z-normalization biases anomaly scores upward in proportion to moving-window variance.** MOMENT uses z-normalization for the normalization and de-normalization steps in the pipeline shown in Figure 1. As a result, the anomaly scores can be decomposed as the product of the variance of the window ($\sigma_i^2$) and the reconstruction MSEs of the normalized window ($T_{i,i+w-1}^{\text{norm}}$). The complete derivation is provided in Appendix D.

---

[2]Twelve of the 23 univariate datasets are derived from multivariate datasets.

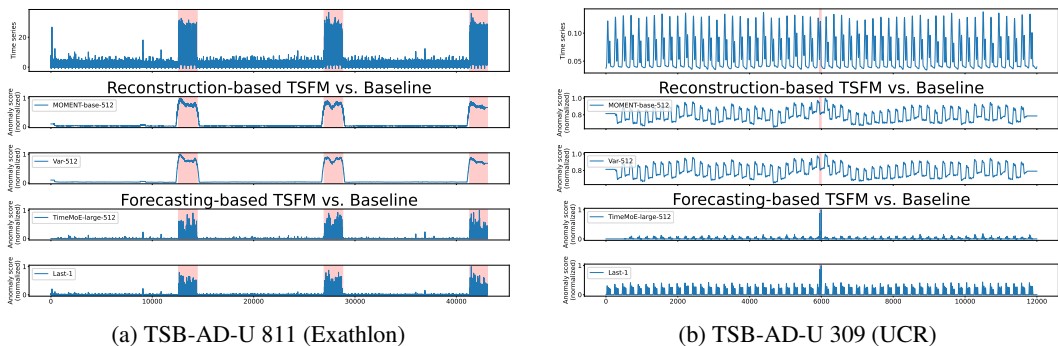

(a) TSB-AD-U 811 (Exathlon)  (b) TSB-AD-U 309 (UCR)

Figure 2: TSFMs and baselines give similar anomaly scores. See Appendix C for more examples.

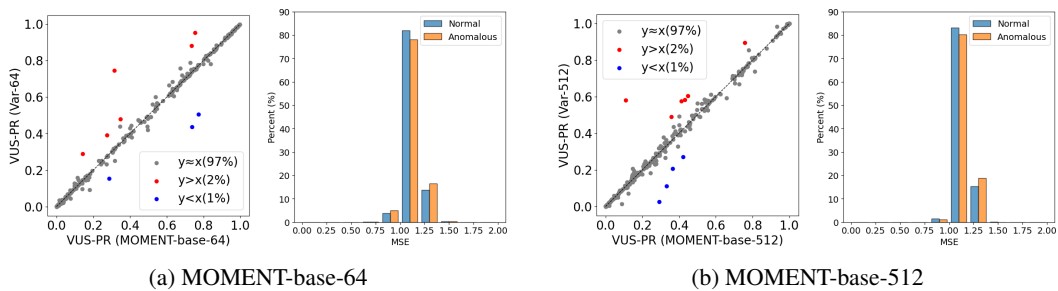

(a) MOMENT-base-64  (b) MOMENT-base-512

Figure 3: One-to-one comparison shows that MOMENT-base and moving-window variance achieve similar VUS-PR scores across the 350 time series in the TSB-AD-U evaluation set (left). The histograms (right) of the reconstruction error, $\text{MSE}\left(T_{i,i+w-1}^{\text{norm}},\ \mathcal{F}^{(\text{reconstruct})}\left(T_{i,i+w-1}^{\text{norm}}\right)\right)$, indicate that this similarity stems from the model's consistently poor reconstruction performance, regardless of whether the context window contains anomaly. See Figure 10 for complete results.

$$s_{i+\lfloor w/2 \rfloor} \approx \sigma_i^2 \cdot \text{MSE}\left(T_{i,i+w-1}^{\text{norm}},\ \mathcal{F}^{(\text{reconstruct})}\left(T_{i,i+w-1}^{\text{norm}}\right)\right) \tag{1}$$

As shown in Figure 3, MOMENT-base reconstructs all[3] normalized windows with a consistently high MSE near 1.1, contradicting the fundamental assumption that anomalies are harder to reconstruct. Consequently, the second factor in Equation 1 exhibits minimal variation, explaining why MOMENT-base performs similarly to moving-window variance for TSAD. An ablation study on the normalization method further consolidates our observation: switching to Min-Max or global z-normalization (normalizing the time series as a whole rather than the individual windows) results in a lower correlation with the baseline (Figure 17).

**Neither larger model size nor larger context window length leads to consistent improvement.** Figure 4 shows that while MOMENT-large yields a more widespread MSE distribution and substantially better reconstructs subsequences than MOMENT-base, the MSE distributions for anomalous and normal windows remain entirely overlapping[4]. This again contradicts the core assumption that anomalies are harder to reconstruct. As a result, although deviating more from the moving-window variance baseline, MOMENT-large still shows no consistent advantage over the latter (see Table 3 and Appendix J for statistical analysis).

## 5.2 FORECASTING-BASED MODELS

**No forecasting-based TSFM significantly outperforms the squared-difference baseline on the task of TSAD.** As illustrated in figure 5, all models achieve similar performance and fail to significantly surpass the squared-difference baseline (Definition 5), regardless of variations in their

---

[3]Constant-value windows are not counted as they are trivially reconstructed.

[4]We demonstrate in Appendix G that this is not a result of data contamination.

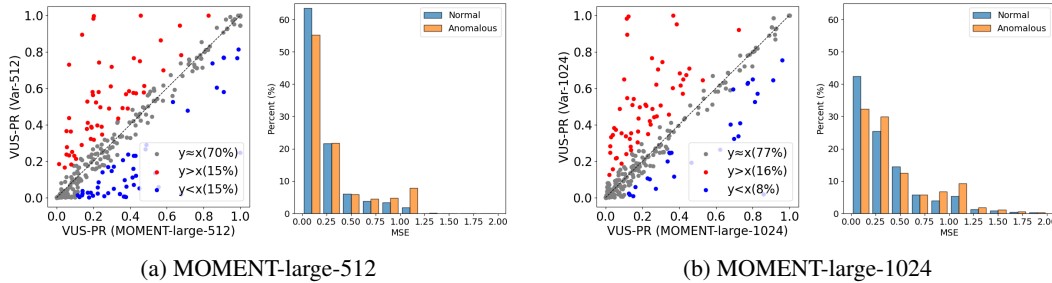

(a) MOMENT-large-512

(b) MOMENT-large-1024

Figure 4: One-to-one comparison shows that MOMENT-large does not consistently outperform moving-window variance across the 350 time series in the TSB-AD-U evaluation set (left). The histograms (right) of the reconstruction error, $\text{MSE}\left(T^{\text{norm}}_{i,i+w-1}, \mathcal{F}^{\text{(reconstruct)}}(T^{\text{norm}}_{i,i+w-1})\right)$, suggest that reconstruction errors are not consistently higher for context windows containing anomalies. See Figure 11 for complete results.

forecasting performance.[5] The statistical analysis shown partly in Table 3 and fully in Appendix J further consolidated this observation. In fact, the VUS-PR scores of all model variants and baselines are also highly correlated with a minimum correlation of 0.9 (Figure 13). In addition, we observed that in some time series, the higher VUS-PR scores achieved by TSFMs stem from issues with the ground truth labeling and the metric's tendency to amplify minor differences. Removing these instances further reduces the performance gap between the baselines and the TSFMs (Figure 16). These secondary findings are detailed in Appendix H.

**Single-step forecasting horizon is the primary limiting factor.** Forecasting-based TSFMs excel at detecting point anomalies but fail to detect sequence anomalies (Liu & Paparrizos, 2024), because the anomaly score is computed using a single forecast for a given context window. For sequence anomalies, the context window overlaps with the anomaly as it slides over the time series, providing the TSFM recent anomalous observations, while the task is to predict just the next *one* observation. As a result, even anomalous values are usually not "unexpected" with regards to the context window, allowing the TSFM to forecast them accurately (see Figure 6). The anomaly is therefore not necessarily harder to predict, contradicting the core assumption of forecasting-based approaches for TSAD. In contrast, when the TSFM forecasts multiple steps ahead from the same context window, the predictions no longer rely on anomalous inputs. The sequence anomaly is then poorly predicted, as shown also in Figure 6, and the TSFM correctly detects it as anomalous. An ablation study on the normalization method offers additional evidence for our observation: the high correlation with the baseline persists when switching to Min-Max, global z-normalization, or mean-scaling (Figure 17). In Section 6.1, we further explore how to exploit long-horizon forecasting.

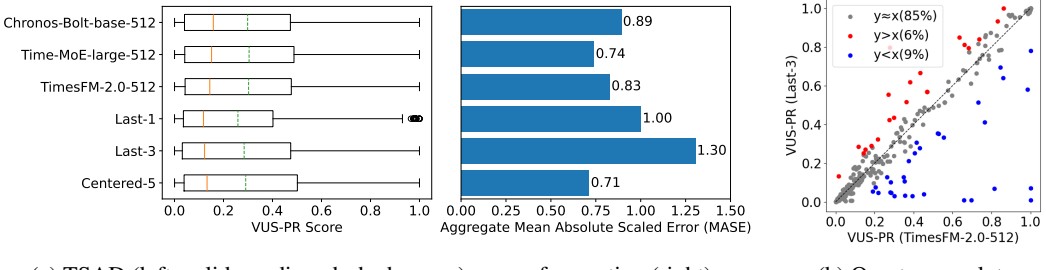

(a) TSAD (left; solid=median, dashed=mean) versus forecasting (right).

(b) One-to-one plot.

Figure 5: Boxplots in (a) show that all TSFM variants achieve similar performance on the TSB-AD-U evaluation set, despite differences in their forecasting performance in the same dataset. Baselines (Last-3 and Centered-5) achieve comparable performance. See Figure 12 for complete results.

---

[5]MASEs are calculated for each series against the Last-1 baseline and aggregated using the geometric mean. For a fair comparison across different context window lengths, evaluation begins at the 513[th] data point.

Table 3: High correlation (with extremely low p-values) and small effect size between the VUS-PR scores of TSFMs and their corresponding baselines (Moving-window variance for Reconstruction-based TSFMs and Last-3 for Forecasting-based TSFMs) over the TSB-AD-U evaluation set. See Appendix J for full results.

|  | mean (TSFM) | mean (baseline) | correlation | p-value | Cohen's $d$ |
|---|---|---|---|---|---|
| **MOMENT-base-512** | 0.311 | 0.313 | 0.991 | 7.91E-308 | -0.047 |
| **MOMENT-large-512** | 0.313 | 0.313 | 0.857 | 1.38E-102 | 0.003 |
| **Chronos-Bolt-base-512** | 0.297 | 0.284 | 0.927 | 1.38e-150 | 0.102 |
| **Time-MoE-large-512** | 0.303 | 0.284 | 0.923 | 8.18e-147 | 0.146 |
| **Times-FM-2.0-512** | 0.302 | 0.284 | 0.923 | 7.82E-147 | 0.142 |

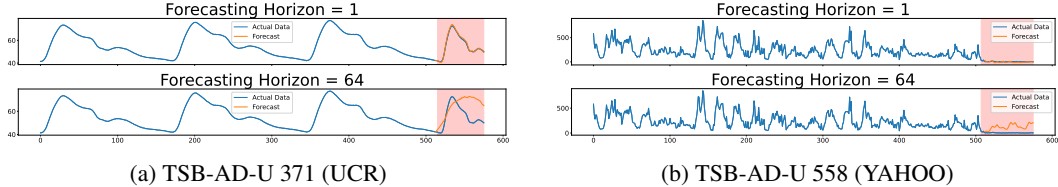

(a) TSB-AD-U 371 (UCR)    (b) TSB-AD-U 558 (YAHOO)

Figure 6: Forecasts from TimesFM-1.0-512 of an anomaly at different forecasting horizons. Errors in anomalous regions are low for a short horizon (1), whereas they are high for a long horizon (64).

## 5.3 TSPULSE

**Zero-shot TSPulse does not outperform moving-window variance and squared-difference.** The reconstruction and forecasting heads (Head$_{time}$ and Head$_{future}$) of TSPulse perform on par with the corresponding baselines. Moreover, the performance of triangulating or ensembling the heads is similar to triangulating or ensembling the two baselines, as shown in Table 4, with TSPulse results taken directly from Ekambaram et al. (2025, Table 1(a)). This holds true even though TSPulse benefits from an additional reconstruction head in the frequency domain (Head$_{fft}$). These observations are not surprising because TSPulse still relies on the conventional reconstruction- and forecasting-based TSAD methodologies (Ekambaram et al., 2025, Section A.8.2), which, as already indicated by our previous experimental results, reduce the potential of all TSFMs to the level of simple one-liners.

Table 4: Mean VUS-PR of TSPulse (zero-shot) vs. baselines on the TSB-AD-U evaluation set

| Reconstruction-based | | Forecasting-based | | Triangulation | Ensemble |
|---|---|---|---|---|---|
| Var-96[†] | 0.42 | Last-3 | 0.28 | 0.46 | 0.43 |
| Var-96[†] | 0.42 | Centered-5 | 0.29 | 0.47 | 0.44 |
| Head$_{time}$ | 0.42 | Head$_{future}$ | 0.3 | 0.48 | 0.44 |

[†]In their evaluation, Ekambaram et al. (2025) calculate reconstruction MSEs on only the last 96 data points of the 512-point context window. Therefore, we use a matching window size of 96.

## 5.4 NON-TSFM ANOMALY DETECTORS

We use the VUS-PR scores obtained in Liu & Paparrizos (2024) for non-TSFM anomaly detectors for an additional comparison with the baselines. **Unlike the TSFMs, the correlation with the baselines is low for the vast majority of non-TSFM anomaly detectors**, as shown partly in Table 5 and fully in Appendix K. Notably, the few exceptions showing relatively high correlation to moving-window variance (Var), e.g., TimesNet (Wu et al., 2023), also employ z-normalization.

## 6 FUTURE DIRECTIONS TO MAKE TSFMS EFFECTIVE FOR TSAD

While our evaluation results are unfavorable to the current application of TSFMs for TSAD, we see alternative directions for improvement. We outline these possibilities in this section.

Table 5: Correlation of VUS-PR scores over the TSB-AD-U evaluation set between non-TSFM TSAD methods and the baselines. See Appendix K for complete results.

|  | correlation with Var | correlation with Last-3 |
|---|---|---|
| **Sub-IForest** (Liu et al., 2008) | 0.301 | 0.279 |
| **Sub-LOF** (Breunig et al., 2000) | -0.088 | 0.140 |
| **MatrixProfile** (Yeh et al., 2016) | -0.034 | 0.242 |
| **KShapeAD** (Paparrizos & Gravano, 2017) | -0.107 | 0.092 |
| **SAND** (Boniol et al., 2021) | -0.063 | 0.097 |
| **Series2Graph** (Boniol & Palpanas, 2020) | 0.464 | 0.338 |
| **KMeansAD** (Yairi et al., 2001) | -0.040 | 0.185 |
| **TimesNet** (Wu et al., 2023) | 0.805 | 0.724 |

## 6.1 COMBINE SHORT AND LONG FORECASTING HORIZONS

In Section 5.2, we identified the single-step forecasting horizon as the primary cause of the poor performance of forecasting-based TSFMs in detecting sequence anomalies. A natural extension – yet largely unexplored in the TSAD literature – is to employ a TSFM that, for a context window $T_{t-w:t-1}$ and forecasting horizon $h$, predicts $\hat{T}_{t:t+h-1}$ and computes the anomaly score of $x_t$ as the mean squared error between $T_{t:t+h-1}$ and $\hat{T}_{t:t+h-1}$. As shown in both individual case studies (Table 6) and aggregate results (Figure 7), increasing $h$ enhances the detection of longer anomalies, but simultaneously reduces sensitivity to shorter anomalies. This trade-off implies that no single fixed horizon $h$ suffices to capture anomalies in various lengths. To address this limitation, we propose an ensemble approach that aggregates anomaly scores across multiple horizons using a MAX-operation. This ensures that observations receive a large anomaly score when they appear anomalous under either a short horizon or a long horizon. Figure 8a illustrates the effect: with $h = 1$, TimesFM-1.0-512 detects the point anomaly but misses the sequence anomaly, whereas with $h = 64$ the reverse occurs. By ensembling both horizons, TimesFM-1.0-512 accurately detects both anomalies.

Table 6: VUS-PR of TimesFM-1.0-512 on TSB-AD-U 371 with a sequence anomaly of length 170 and TSB-AD-U 603 with a point anomaly for increasing forecasting horizons $h$.

| ID | Anomaly | $h = 1$ | $h = 2$ | $h = 4$ | $h = 8$ | $h = 16$ | $h = 32$ | $h = 64$ | $h = 128$ |
|---|---|---|---|---|---|---|---|---|---|
| 371 | Sequence | 0.04 | 0.1 | 0.21 | 0.27 | 0.36 | 0.46 | 0.65 | 0.68 |
| 603 | Point | 1.0 | 0.8 | 0.9 | 0.94 | 0.9 | 0.47 | 0.27 | 0.02 |

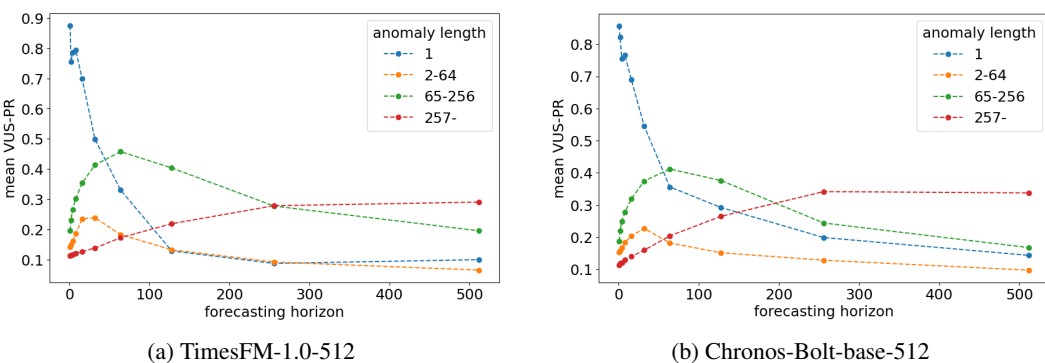

(a) TimesFM-1.0-512           (b) Chronos-Bolt-base-512

Figure 7: Mean VUS-PR scores with various forecasting horizons, evaluated on time series containing anomalies of different lengths within the TSB-AD-U evaluation set. See Appendix L for complete results.

## 6.2 DETECT ANOMALIES IN THE HIDDEN REPRESENTATIONS

The hidden representations of TSFMs capture meaningful time series characteristics such as trend, amplitude, frequency, and phase (Goswami et al., 2024). These representations can be directly

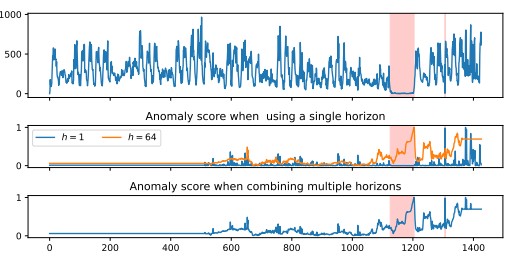 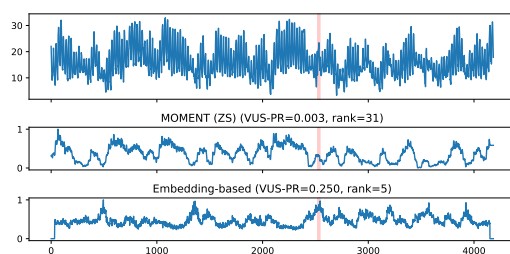

(a) TSB-AD-U 558 (Yahoo) with a sequence and point anomaly. TimesFM with $h = 1$ or $h = 64$ fails to detect both anomalies, while their ensemble succeeds.

(b) TSB-AD-U 474 (UCR). A k-NN model using the learned representation of MOMENT significantly outperforms reconstruction-based MOMENT.

Figure 8: Qualitative examples of our suggestions to enhance the performance of TSFMs on TSAD.

leveraged for anomaly detection, as illustrated in Figure 8b. We use MOMENT because it provides direct access to hidden representations, employing the base model with a window size of 64 as Liu & Paparrizos (2024). When anomaly scores are based on reconstruction error, MOMENT achieves a VUS-PR of only 0.003 and ranks 31st out of 32 detectors. In contrast, applying KNN to the hidden representations effectively detects the anomaly, achieving a significantly higher VUS-PR of 0.250 and improving the rank to 5th in the benchmark of Liu & Paparrizos (2024). Although other detectors (e.g., KMeansAD) achieve even higher performance, this example highlights the potential of TSFM-derived representations to enhance anomaly detection.

## 6.3 FINE-TUNE TSFMs FOR TSAD USING LABELED ANOMALIES

Foundation models are trained with forecasting or reconstruction objectives (Definitions 2 and 3). An intrinsic problem with this approach is that the training objective is not aligned with detecting anomalies. As a result, foundation models lack explicit knowledge of what constitutes an anomaly. This issue is further amplified by the fact that anomalies heavily depend on the domain. Further fine-tuning on the pre-training task does not address this misalignment, as it only enhances forecasting or reconstruction performance rather than the models' anomaly detection capabilities. We therefore argue that effective TSAD with TSFMs may benefit from end-to-end fine-tuning of the foundation models using labeled anomalies. This is analogous to how pre-trained LLMs are fine-tuned for sequence labeling tasks specifically, such as part-of-speech tagging or named entity recognition (Devlin et al., 2019; Tjong Kim Sang & De Meulder, 2003). Even more, TSFMs potentially only require a limited number of anomalous events to adapt their learned representations of time series to new domains. This is especially important for TSAD, in which it is often costly to collect large sets of labeled anomalies (Chandola et al., 2009). A comprehensive study of this paradigm is beyond the scope of the present work, but we highlight it as a promising direction for future research.

## 7 CONCLUSION

In this work, we showed that the current approaches to apply TSFMs to TSAD are *not* effective. Our analysis shows that anomalies are not consistently harder to reconstruct or forecast with TSFMs, undermining the key assumptions on which existing methods rely. These findings hold irrespective of model size, context window length, or overall reconstruction/forecasting performance. Specifically, we found no significant difference between TSFMs and two simple baselines that were selected to mimic their predictions for TSAD. We also emphasized the importance of visually analyzing both data and algorithm outputs, because aggregated evaluation metrics often hide the true strengths and weaknesses of an algorithm. Looking forward, we suggest three alternative directions to leverage the potential of TSFMs in TSAD: (1) ensemble short and long forecasting horizons, (2) detect anomalies in the hidden representations, and (3) fine-tune using labeled anomalies.

ACKNOWLEDGMENTS

This research is supported by the Flemish government under the Flanders AI Research Program, by Flanders Make, the strategic research centre for the manufacturing industry, and by Internal Funds KU Leuven (C3/23/047). This work used the resources and services provided by the VSC (Flemish Supercomputer Center), funded by the Research Foundation - Flanders (FWO) and the Flemish Government.

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

# A  DECLARATIONS

## A.1  USE OF LARGE LANGUAGE MODELS

We used Large Language Models (LLMs) exclusively for polishing the text. All research ideation, literature review, experimental design, implementations and analysis were carried out by the authors. The LLMs did not generate any scientific content or influence the scientific findings in this manuscript.

## A.2  CODE AND DATA AVAILABILITY

1. **Code.** All our scripts to load the time series, detect anomalies, and evaluate the performances are publicly available at `https://gitlab.kuleuven.be/m-group-campus-brugge/dtai_public/publications/iclr2026_timeseriesfoundationmodelsad/`. Additionally, we relied on code from:

   TSB-AD (`https://github.com/TheDatumOrg/TSB-AD`), MOMENT (`https://github.com/moment-timeseries-foundation-model/moment`), TimesFM (`https://github.com/google-research/timesfm`), Chronos (`https://github.com/amazon-science/chronos-forecasting`), Time-MoE (`https://github.com/Time-MoE/Time-MoE`), and TSPulse (`https://github.com/ibm-granite/granite-tsfm`).

2. **Experimental results.** We added the experimental results in supplementary material, but these are also available at `https://gitlab.kuleuven.be/m-group-campus-brugge/dtai_public/publications/iclr2026_timeseriesfoundationmodelsad/`. In addition, we also have the raw anomaly scores, reconstructed subsequences, and model forecasts available in the online repository.

3. **Data.** We relied on publicly available data (Liu & Paparrizos, 2024). The raw data files are available at `https://github.com/TheDatumOrg/TSB-AD?tab=readme-ov-file#dataset` and the train-test split at `https://github.com/TheDatumOrg/TSB-AD/tree/main/Datasets/File_List/`.

# B    DESIGN CHOICES FOR TSFMS

We briefly introduce variations in the design choices for TSFMs, focusing on the four Transformer-based families examined in our experiments: MOMENT (Goswami et al., 2024), Chronos (Ansari et al., 2024a;b), TimesFM (Das et al., 2024), and Time-MoE (Shi et al., 2024).

## B.1    NORMALIZATION

One difference between natural language and time series is that while the number of words in a language is finite, a data point in a time series can, in principle, take on infinite, unbounded values. To handle any time series, TSFMs include a normalization step during preprocessing and a mirroring de-normalization step in postprocessing.

In this regard, Time-MoE (Shi et al., 2024), MOMENT (Goswami et al., 2024), Chronos-Bolt (Ansari et al., 2024b), and TimesFM (Das et al., 2024) all use z-score normalization, which typically transforms each context window into a series with a mean of 0 and a standard deviation of 1. Notably, Chronos-t5 (Ansari et al., 2024a), the precursor of Chronos-Bolt, employs mean scaling, which divides each data point in the context window by the mean of the window.

## B.2    TOKENIZATION & EMBEDDING

For LLMs, the process of converting a sequence of text into a sequence of embeddings consists of two steps. First, the sequence is broken down into smaller, discrete units called tokens, which can be words or subwords, in the tokenization step. The embedding step then converts these tokens into vectors called embeddings, which capture both syntactic and semantic information of the tokens.

Among popular families of Transformer-based TSFMs, only Chronos-t5 employs discrete tokens like LLMs. In Chronos-t5, each data point is mapped to a discrete token through uniform binning, with an embedding learned for each token. In contrast, Time-MoE directly maps each data point to an embedding using a learned SwiGLU layer (Shazeer, 2020), while MOMENT, TimesFM, and Chronos-Bolt segment data points into equal-length subsequences called patches, which are then mapped to embeddings via a learned linear projection or a residual block.

## B.3    MODEL ARCHITECTURE

The vast majority of LLMs are based on variants of the transformer architecture introduced in 2017 (Vaswani et al., 2017). A transformer typically transforms a sequence of input embeddings into another sequence of output embeddings, enabling different tasks such as next-token prediction or sequence reconstruction. The original transformer architecture included both an encoder and a decoder. However, architectures featuring only an encoder (Devlin et al., 2019) or only a decoder (Radford et al., 2018) have since been proposed, with the decoder-only architecture becoming the de facto choice for the latest LLMs.

The four families of Transformer-based TSFMs examined in our experiments fall into the three categories of model architecture mentioned above. Chronos uses an encoder-decoder architecture, MOMENT uses an encoder-only architecture, and Time-MoE and TimesFM both use a decoder-only architecture.

## B.4    PRE-TRAINING TASK

Before being adapted to various downstream tasks, foundation models are pre-trained on a single task using a huge amount of training data in a self-supervised manner. For this pre-training task, Chronos, Time-MoE, and TimesFM, like the vast majority of current TSFMs, all use forecasting. More specifically, for the loss function in pre-training, Chronos uses the cross entropy loss for the next data point, TimesFM uses the Mean Squared Error for the next patch, while Time-MoE involves forecasting with different prediction horizons. On the other hand, MOMENT is the only large-scale TSFM, as far as we know, that is pre-trained on the task of sequence reconstruction.

# C    MORE EXAMPLES ON THE SIMILARITY BETWEEN TSFMS AND BASELINES

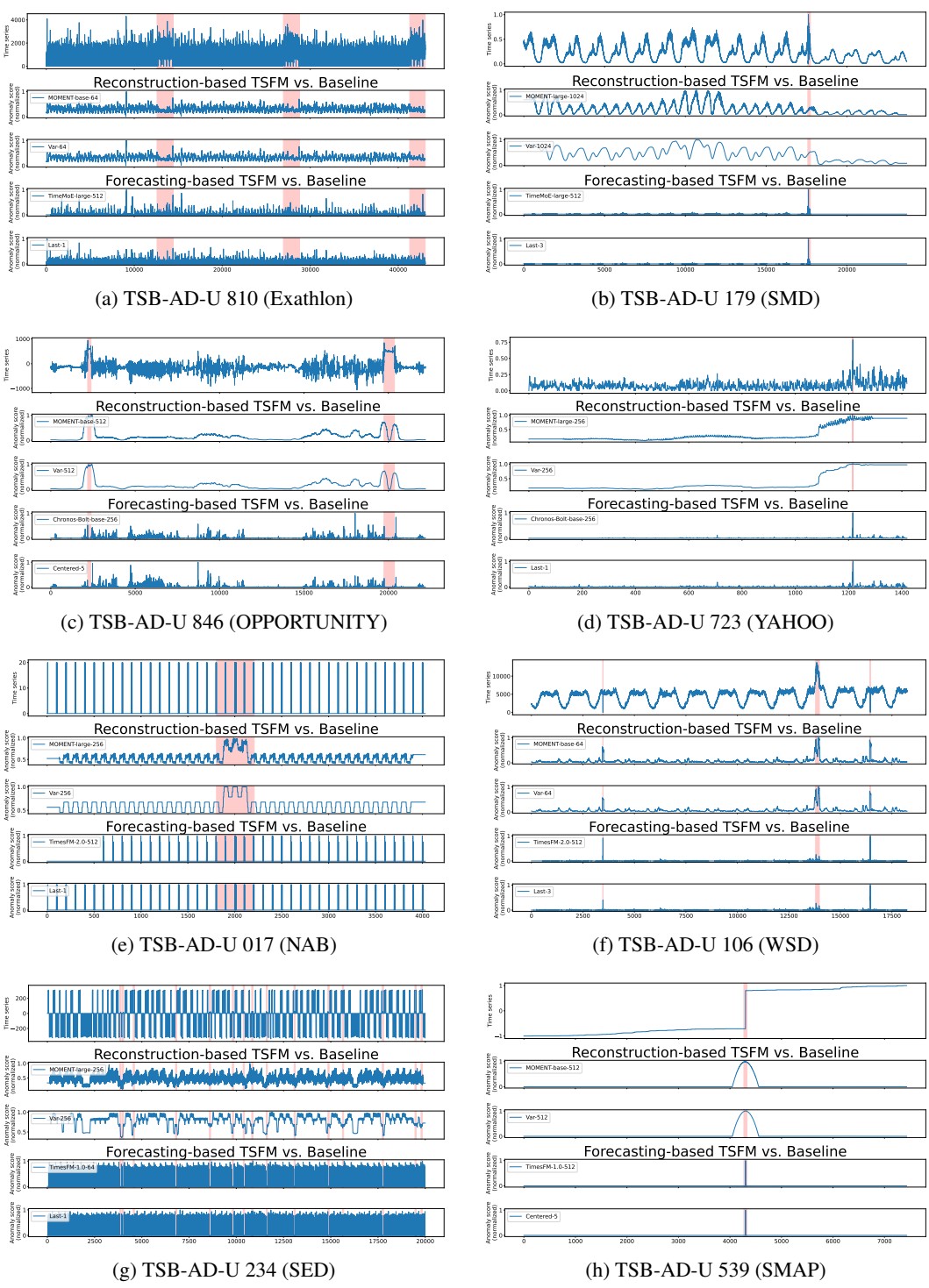

Figure 9: TSFMs and baselines give similar anomaly scores.

# D DERIVATION FOR THE DECOMPOSITION OF ANOMALY SCORES GIVEN BY MOMENT

MOMENT uses z-normalization for the normalization and de-normalization steps in the pipelines shown in Figure 1. As a result, the anomaly scores can be decomposed as the product of the variance of the window and the reconstruction MSEs of the normalized window. The derivation for this conclusion is as below, where $T^{norm}_{i,i+w-1}$ denotes the normalized context window, which is fed into the TSFM $\mathcal{F}^{(\text{reconstruct})}$ for reconstruction.

$$\mu_i := \text{mean}\left(T_{i,i+w-1}\right)$$
$$\sigma_i := \text{std}\left(T_{i,i+w-1}\right)$$
$$s_{i+\lfloor w/2 \rfloor} = \text{MSE}\left(T_{i,i+w-1},\ \hat{T}_{i,i+w-1}\right)$$
$$= \text{MSE}\left(T^{norm}_{i,i+w-1} \cdot (\sigma_i + \epsilon) + \mu_i,\ \mathcal{F}^{(\text{reconstruct})}(T^{norm}_{i,i+w-1}) \cdot (\sigma_i + \epsilon) + \mu_i\right)$$
$$= \text{MSE}\left(T^{norm}_{i,i+w-1} \cdot (\sigma_i + \epsilon),\ \mathcal{F}^{(\text{reconstruct})}(T^{norm}_{i,i+w-1}) \cdot (\sigma_i + \epsilon)\right)$$
$$= (\sigma_i + \epsilon)^2 \cdot \text{MSE}\left(T^{norm}_{i,i+w-1},\ \mathcal{F}^{(\text{reconstruct})}(T^{norm}_{i,i+w-1})\right)$$
$$\approx \sigma_i^2 \cdot \text{MSE}\left(T^{norm}_{i,i+w-1},\ \mathcal{F}^{(\text{reconstruct})}(T^{norm}_{i,i+w-1})\right)$$

Note that MOMENT's built-in normalizer adds a small $\epsilon = 10^{-5}$ to the true standard deviation to avoid division by zero, but its influence on our conclusion here is negligible.

# E  COMPLETE EXPERIMENTAL RESULTS FOR RECONSTRUCTION-BASED MODELS

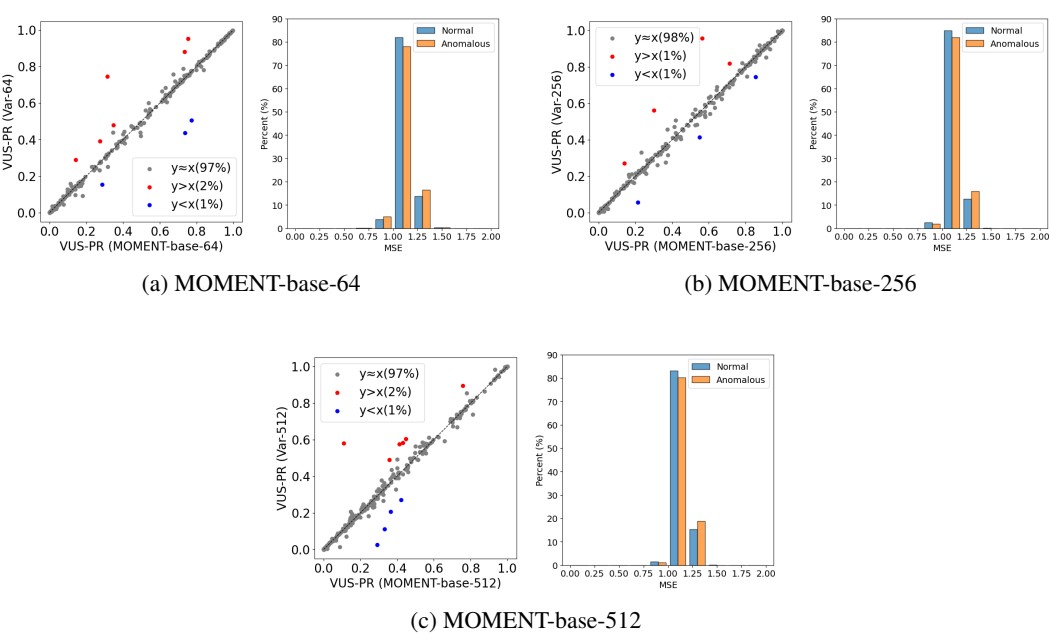

(a) MOMENT-base-64

(b) MOMENT-base-256

(c) MOMENT-base-512

Figure 10: One-to-one comparison between MOMENT-base and moving-window variance across 350 time series in the TSB-AD-U evaluation set (left); Distribution of reconstruction MSEs of the normalized context windows (right)

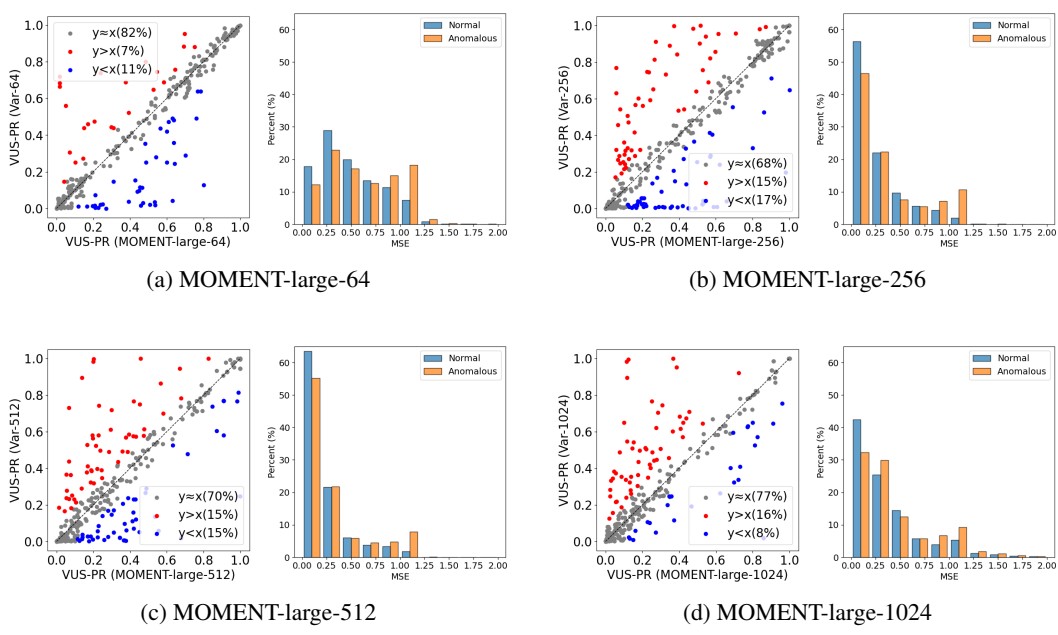

(a) MOMENT-large-64

(b) MOMENT-large-256

(c) MOMENT-large-512

(d) MOMENT-large-1024

Figure 11: One-to-one comparison between MOMENT-large and moving-window variance across 350 time series in the TSB-AD-U evaluation set (left); Distribution of reconstruction MSEs of the normalized context windows (right)

# F  Complete experimental results for Forecasting-based Models

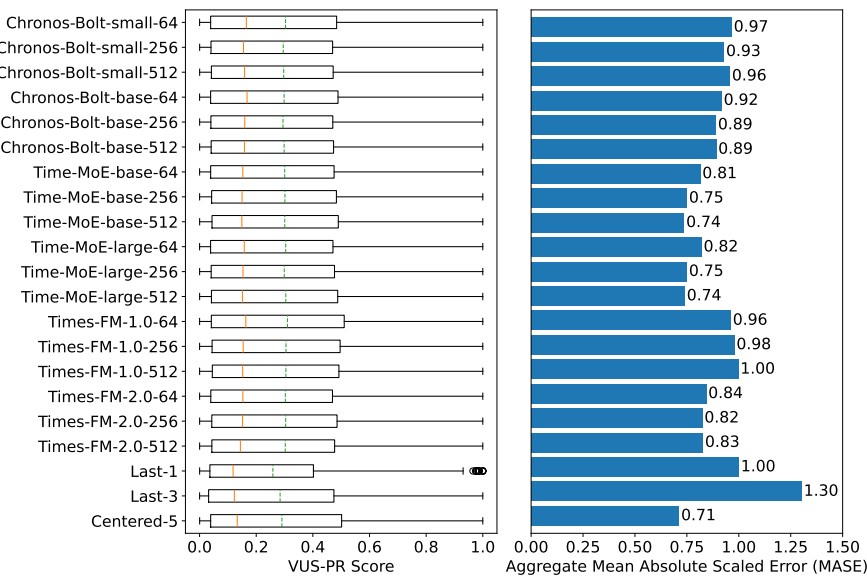

Figure 12: Boxplots (solid=median, dashed=mean) show that all TSFM variants achieve similar performance on the TSB-AD-U evaluation set, despite differences in their forecasting performance in the same dataset. Baselines (Last-3 and Centered-5) achieve performance comparable with TSFMs.

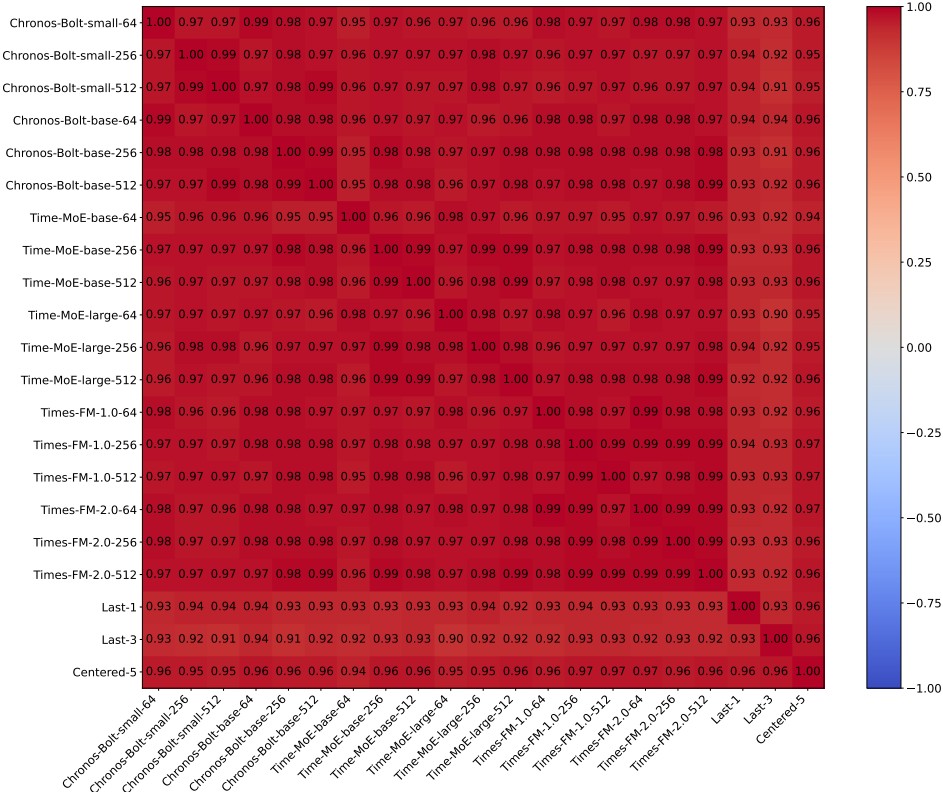

Figure 13: All model variants and baselines show high pairwise correlation in their VUS-PR scores across 350 time series in TSB-AD-U, with the minimum correlation being 0.90.

# G    DATA CONTAMINATION FOR MOMENT

While none of the forecasting-based TSFMs covered in our experiments was pre-trained on TSAD datasets (Ansari et al., 2024a; Shi et al., 2024; Das et al., 2024), the only reconstruction-based one, MOMENT, did use some TSAD datasets in its pre-training (Goswami et al., 2024). One possible explanation for our observation that MOMENT reconstructs windows containing anomalies equally well (or poorly) when compared to windows containing no anomalies is data contamination. As shown in Figure 14, our observation still holds with time series coming from datasets[6] not used in MOMENT's pre-training, refuting this explanation.

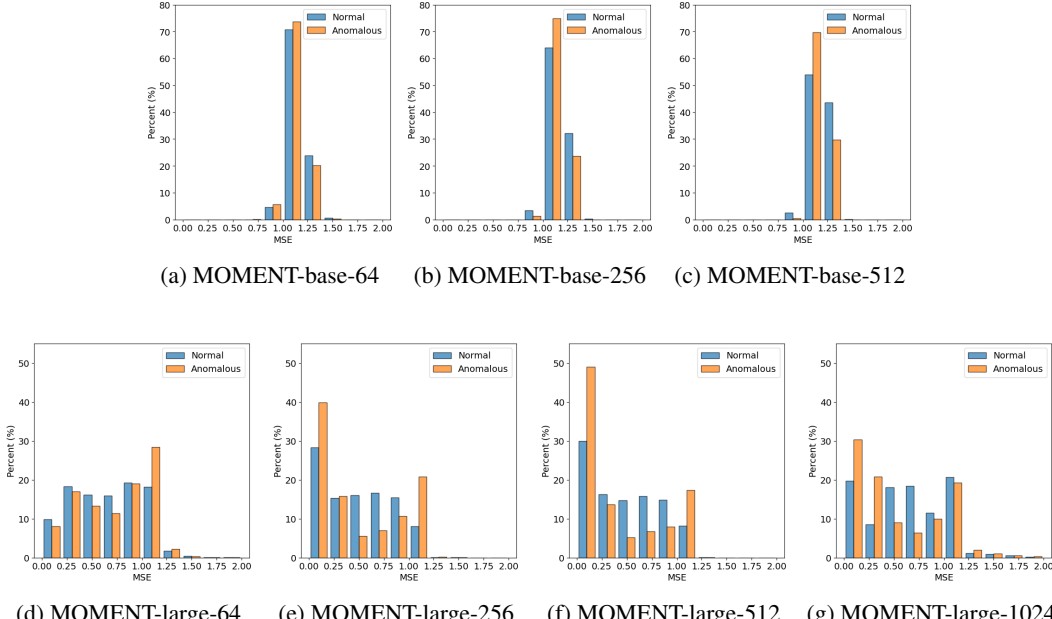

(a) MOMENT-base-64    (b) MOMENT-base-256    (c) MOMENT-base-512

(d) MOMENT-large-64    (e) MOMENT-large-256    (f) MOMENT-large-512    (g) MOMENT-large-1024

Figure 14: Distribution of reconstruction MSEs of the normalized context windows with 94 time series in the TSB-AD-U evaluation set, coming from datasets not used in MOMENT's pre-training

---

[6]WSD, Stock, LTDB, SED, TAO, NEK, CATSv2, TODS, Power, SWaT, Exathlon

## H    SECONDARY FINDINGS

We observed that in some time series, the higher VUS-PR scores achieved by forecasting-based TSFMs, when compared with the squared-difference baseline, stem from issues with the ground truth labeling and the metric's tendency to amplify minor differences. We illustrate this with a representative case.

Figure 15 shows a time series from the TSB-AD-U evaluation set. All data points in this time series are equal to -1, except for one point which is equal to 1. The ground truth indicates that an anomaly happened from 16 time steps before this deviating value, and lasted until 34 steps after it. For this series, TimeMoE-large-512 got a VUS-PR score of 0.51 while the score for the Last-1 baseline is only 0.05. Figure 15 reveals the following problems:

- **Problematic labeling**: it is hard to understand why the data points around the spike should also be labeled as anomalous. In particular, why should there be more data points marked as anomalous after the spike than before the spike?

- **Exaggerating metric**: given the ground truth, it is hard to tell why TimeMoE should get a much higher performance score than the Last-1 baseline when looking at their anomaly scores plotted on a linear scale (the second and the third subfigures). The reason only becomes visible when the anomaly scores from TimeMoE are plotted on a logarithmic scale (the fourth subfigure). This reveals tiny bumps at the level of $10^{-5}$, just to the right of the spike, where Last-1, in contrast, yields exact zeros. Because VUS-PR (Paparrizos et al., 2022), like its parent AUC-PR, only cares about the relative ranking of the scores rather than their actual values, these tiny bumps result in a significantly higher VUS-PR for TimeMoE.

Our inspection identified 50 time series with problematic labeling in the TSB-AD-U evaluation set. The indices of the 50 problematic time series are listed in Table 7. For 47 of them, the labeling issue involves anomalous labels (often arbitrarily) overlapping with normal regions, similar to the example shown in Figure 15. The remaining three, from the UCR dataset, exhibit off-by-one mislabeling.

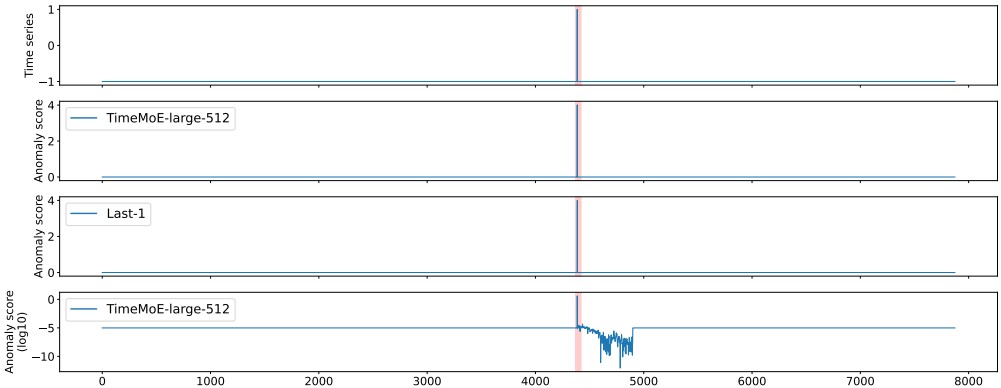

Figure 15: Case study with TSB-AD-U 534 (SMAP)

| Dataset | Index | Problem |
|---|---|---|
| NAB | 0–22 | |
| SMAP | 245–261 | |
| SWAT | 262 | Anomalous labeling overlapping with normal regions |
| MSL | 43, 45, 47, 48 | |
| SMD | 68, 74 | |
| UCR | 212, 221, 228 | Off-by-one mislabeling |

Table 7: Time series with problematic labeling in the TSB-AD-U evaluation set

Removing these instances further reduced the performance gap between forecasting-based TSFMs and the squared-difference baseline, as shown in Figure 16.

**Without Problematic Instances**

**With Problematic Instances**

Figure 16: One-to-one comparison between TimesFM and the squared-difference baseline over the 350 time series in the TSB-AD-U evaluation set, with and without time series with problematic labeling.

# I ABLATION STUDY ON NORMALIZATION

**Reconstruction-based Models**

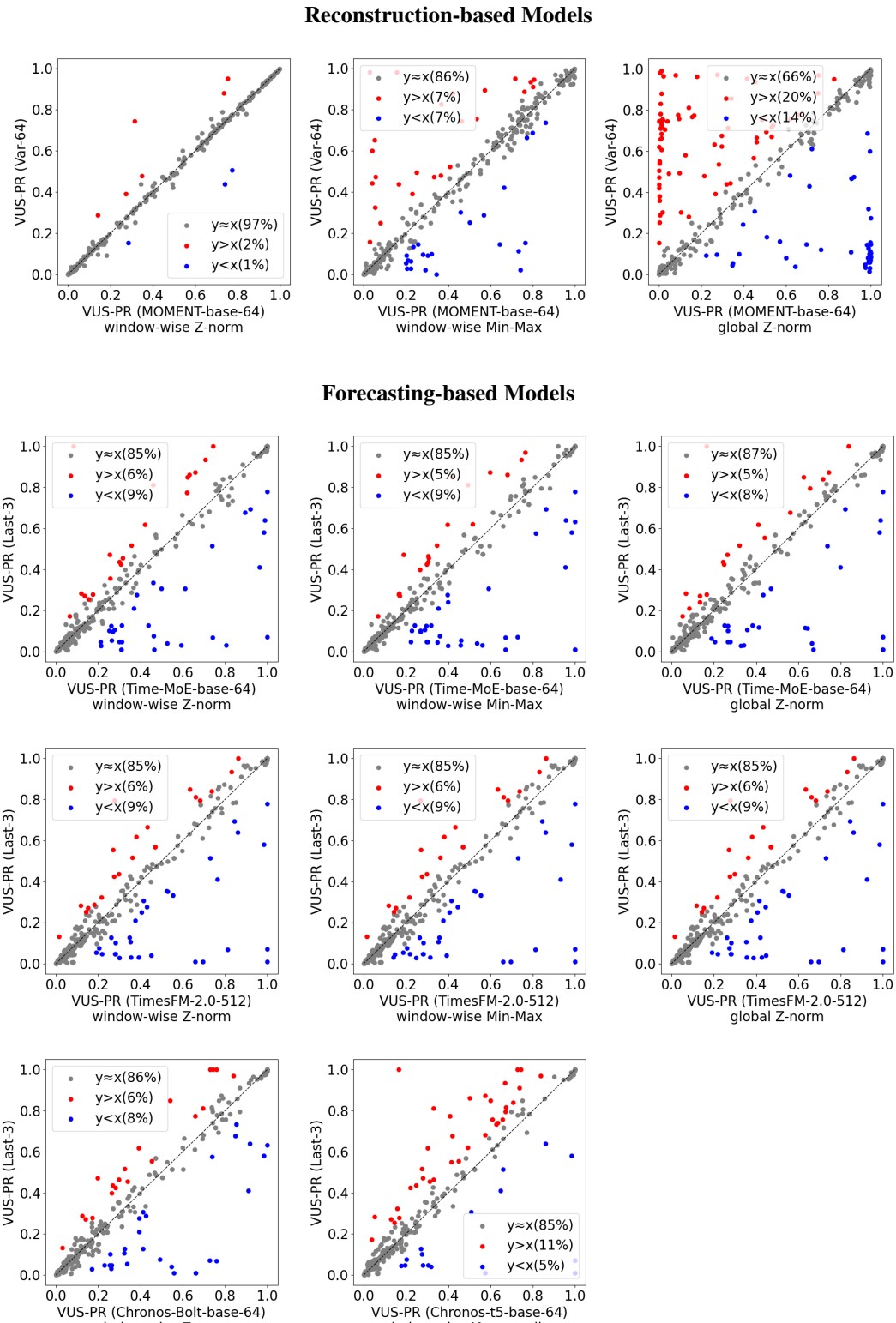

**Forecasting-based Models**

Figure 17: When employing normalization methods other than window-wise Z-norm, the correlation with baselines diminishes for reconstruction-based models but persists for forecasting-based models.

## J  STATISTICAL ANALYSIS

Table 8: High correlation (with extremely low p-values) and small effect size between the VUS-PR scores of TSFMs and their corresponding baselines (Moving-window variance for Reconstruction-based TSFMs and Last-3 for Forecasting-based TSFMs) over the TSB-AD-U evaluation set.

| | mean (TSFM) | mean (baseline) | correlation | p-value | Cohen's $d$ |
|---|---|---|---|---|---|
| **MOMENT-base-64** | 0.412 | 0.414 | 0.994 | 0 | -0.048 |
| **MOMENT-base-256** | 0.386 | 0.386 | 0.995 | 0 | -0.005 |
| **MOMENT-base-512** | 0.311 | 0.313 | 0.991 | 7.91E-308 | -0.047 |
| **MOMENT-large-64** | 0.427 | 0.414 | 0.926 | 2.90E-149 | 0.091 |
| **MOMENT-large-256** | 0.389 | 0.386 | 0.848 | 1.87E-98 | 0.016 |
| **MOMENT-large-512** | 0.313 | 0.313 | 0.857 | 1.38E-102 | 0.003 |
| **MOMENT-large-1024** | 0.245 | 0.273 | 0.855 | 4.78E-101 | -0.180 |
| **Chronos-Bolt-small-64** | 0.304 | 0.284 | 0.926 | 1.45e-149 | 0.153 |
| **Chronos-Bolt-small-256** | 0.294 | 0.284 | 0.920 | 7.27e-144 | 0.077 |
| **Chronos-Bolt-small-512** | 0.295 | 0.284 | 0.914 | 6.71e-139 | 0.081 |
| **Chronos-Bolt-base-64** | 0.299 | 0.284 | 0.937 | 1.29e-161 | 0.124 |
| **Chronos-Bolt-base-256** | 0.293 | 0.284 | 0.916 | 9.76e-141 | 0.068 |
| **Chronos-Bolt-base-512** | 0.297 | 0.284 | 0.927 | 1.38e-150 | 0.102 |
| **Time-MoE-base-64** | 0.300 | 0.284 | 0.918 | 9.82e-142 | 0.121 |
| **Time-MoE-base-256** | 0.300 | 0.284 | 0.932 | 1.41e-155 | 0.132 |
| **Time-MoE-base-512** | 0.299 | 0.284 | 0.932 | 1.07e-155 | 0.124 |
| **Time-MoE-large-64** | 0.304 | 0.284 | 0.899 | 3.00e-127 | 0.134 |
| **Time-MoE-large-256** | 0.297 | 0.284 | 0.918 | 7.17e-142 | 0.095 |
| **Time-MoE-large-512** | 0.303 | 0.284 | 0.923 | 8.18e-147 | 0.146 |
| **Times-FM-1.0-64** | 0.310 | 0.284 | 0.918 | 5.14E-142 | 0.194 |
| **Times-FM-1.0-256** | 0.305 | 0.284 | 0.934 | 8.05E-158 | 0.175 |
| **Times-FM-1.0-512** | 0.305 | 0.284 | 0.933 | 3.77E-157 | 0.172 |
| **Times-FM-2.0-64** | 0.303 | 0.284 | 0.923 | 9.70E-147 | 0.144 |
| **Times-FM-2.0-256** | 0.304 | 0.284 | 0.926 | 1.56E-149 | 0.156 |
| **Times-FM-2.0-512** | 0.302 | 0.284 | 0.923 | 7.82E-147 | 0.142 |

On top of the results shown in Table 8 above, we also conducted non-inferiority tests based on Wilcoxon signed-rank, with which we can exclude the possibility (with at least 95% confidence level) that the TSFMs outperform the corresponding baselines by more than a small margin (0.016 for MOMENT-large, 0.0008 for MOMENT-base, and 0.008 for all forecasting-based TSFMs).

Moreover, our key findings still hold when the metric is switched from VUS-PR to Event-based F1, as shown in Table 9 below. In parallel, for Event-based F1, the same non-inferiority tests show that we can exclude the possibility (with at least 95% confidence level) that the TSFMs outperform the corresponding baselines by more than a small margin (0.025 for MOMENT-large, 0.0001 for MOMENT-base, and 0.02 for all forecasting-based TSFMs).

Table 9: High correlation (with extremely low p-values) and small effect size between the Event-based F1 scores of TSFMs and their corresponding baselines (Moving-window variance for Reconstruction-based TSFMs and Last-3 for Forecasting-based TSFMs) over the TSB-AD-U evaluation set.

| | mean (TSFM) | mean (baseline) | correlation | p-value | Cohen's $d$ |
|---|---|---|---|---|---|
| **MOMENT-base-64** | 0.495 | 0.487 | 0.985 | 3.29e-270 | 0.118 |
| **MOMENT-base-256** | 0.419 | 0.417 | 0.996 | 0 | 0.065 |
| **MOMENT-base-512** | 0.345 | 0.345 | 0.994 | 0 | 0.008 |
| **MOMENT-large-64** | 0.509 | 0.487 | 0.864 | 3.76e-106 | 0.106 |
| **MOMENT-large-256** | 0.465 | 0.417 | 0.794 | 1.24e-77 | 0.199 |
| **MOMENT-large-512** | 0.383 | 0.345 | 0.809 | 1.95e-82 | 0.179 |
| **MOMENT-large-1024** | 0.287 | 0.300 | 0.866 | 1.11e-106 | -0.077 |
| **Chronos-Bolt-small-64** | 0.633 | 0.619 | 0.840 | 6.36e-95 | 0.062 |
| **Chronos-Bolt-small-256** | 0.636 | 0.619 | 0.871 | 6.89e-110 | 0.088 |
| **Chronos-Bolt-small-512** | 0.625 | 0.619 | 0.845 | 9.84e-97 | 0.028 |
| **Chronos-Bolt-base-64** | 0.623 | 0.619 | 0.900 | 4.19e-128 | 0.022 |
| **Chronos-Bolt-base-256** | 0.626 | 0.619 | 0.871 | 5.89e-110 | 0.034 |
| **Chronos-Bolt-base-512** | 0.621 | 0.619 | 0.868 | 5.29e-108 | 0.009 |
| **Time-MoE-base-64** | 0.626 | 0.619 | 0.884 | 2.22e-117 | 0.039 |
| **Time-MoE-base-256** | 0.650 | 0.619 | 0.881 | 1.91e-115 | 0.162 |
| **Time-MoE-base-512** | 0.650 | 0.619 | 0.859 | 9.73e-104 | 0.147 |
| **Time-MoE-large-64** | 0.635 | 0.619 | 0.887 | 5.99e-119 | 0.084 |
| **Time-MoE-large-256** | 0.653 | 0.619 | 0.877 | 3.46e-113 | 0.172 |
| **Time-MoE-large-512** | 0.660 | 0.619 | 0.855 | 1.45e-101 | 0.193 |
| **Times-FM-1.0-64** | 0.632 | 0.619 | 0.853 | 2.04e-100 | 0.063 |
| **Times-FM-1.0-256** | 0.644 | 0.619 | 0.866 | 3.87e-107 | 0.125 |
| **Times-FM-1.0-512** | 0.638 | 0.619 | 0.880 | 4.23e-115 | 0.097 |
| **Times-FM-2.0-64** | 0.631 | 0.619 | 0.880 | 5.67e-115 | 0.062 |
| **Times-FM-2.0-256** | 0.636 | 0.619 | 0.860 | 3.14e-104 | 0.080 |
| **Times-FM-2.0-512** | 0.633 | 0.619 | 0.860 | 3.51e-104 | 0.067 |

## K    COMPARISON WITH NON-TSFM TSAD METHODS

Table 10: Correlation of VUS-PR scores over the TSB-AD-U evaluation set between non-TSFM TSAD methods and the baselines. The scores of the non-TSFM TSAD methods are directly taken from Liu & Paparrizos (2024). The window size of the Moving-window Variance baseline (Var) is chosen to be the same with the window size of the TSAD method in comparison. When the TSAD method does not involve a window size, it is compared with Var-64.

|  | correlation with Var | correlation with Last-3 |
|---|---|---|
| **Sub-IForest** (Liu et al., 2008) | 0.301 | 0.279 |
| **IForest** (Liu et al., 2008) | 0.534 | 0.628 |
| **Sub-LOF** (Breunig et al., 2000) | -0.088 | 0.140 |
| **LOF** (Breunig et al., 2000) | 0.395 | 0.654 |
| **POLY** (Li et al., 2007) | 0.795 | 0.436 |
| **MatrixProfile** (Yeh et al., 2016) | -0.034 | 0.242 |
| **KShapeAD** (Paparrizos & Gravano, 2017) | -0.107 | 0.092 |
| **SAND** (Boniol et al., 2021) | -0.063 | 0.097 |
| **Series2Graph** (Boniol & Palpanas, 2020) | 0.464 | 0.338 |
| **SR** (Ren et al., 2019) | 0.556 | 0.805 |
| **Sub-PCA** (Aggarwal, 2017) | 0.470 | 0.203 |
| **Sub-HBOS** (Goldstein & Dengel, 2012) | 0.285 | 0.250 |
| **Sub-OCSVM** (Schölkopf et al., 1999) | 0.104 | 0.213 |
| **Sub-MCD** (Rousseeuw & Driessen, 1999) | -0.046 | 0.293 |
| **Sub-KNN** (Ramaswamy et al., 2000) | -0.045 | 0.075 |
| **KMeansAD** (Yairi et al., 2001) | -0.040 | 0.185 |
| **AutoEncoder** (Sakurada & Yairi, 2014) | 0.378 | 0.361 |
| **CNN** (Munir et al., 2018) | 0.660 | 0.704 |
| **LSTMAD** (Malhotra et al., 2015) | 0.518 | 0.587 |
| **TranAD** (Tuli et al., 2022) | 0.472 | 0.464 |
| **AnomalyTransformer** (Xu et al., 2022) | 0.415 | 0.491 |
| **OmniAnomaly** (Su et al., 2019) | 0.494 | 0.435 |
| **USAD** (Audibert et al., 2020) | 0.615 | 0.277 |
| **Donut** (Xu et al., 2018) | 0.580 | 0.527 |
| **TimesNet** (Wu et al., 2023) | 0.805 | 0.724 |
| **FITS** (Xu et al., 2024) | 0.773 | 0.621 |

## L    FORECASTING HORIZONS FOR FORECASTING-BASED TSFMS

As depicted in Figure 18, under the forecasting-based methodology, increasing forecasting horizon can improve TSFMs' TSAD performance for sequence anomalies, at the expense of performance deterioration for point anomalies. Furthermore, the optimal forecasting horizon correlates with the anomaly length, as evidenced by the peaks in Figure 18.

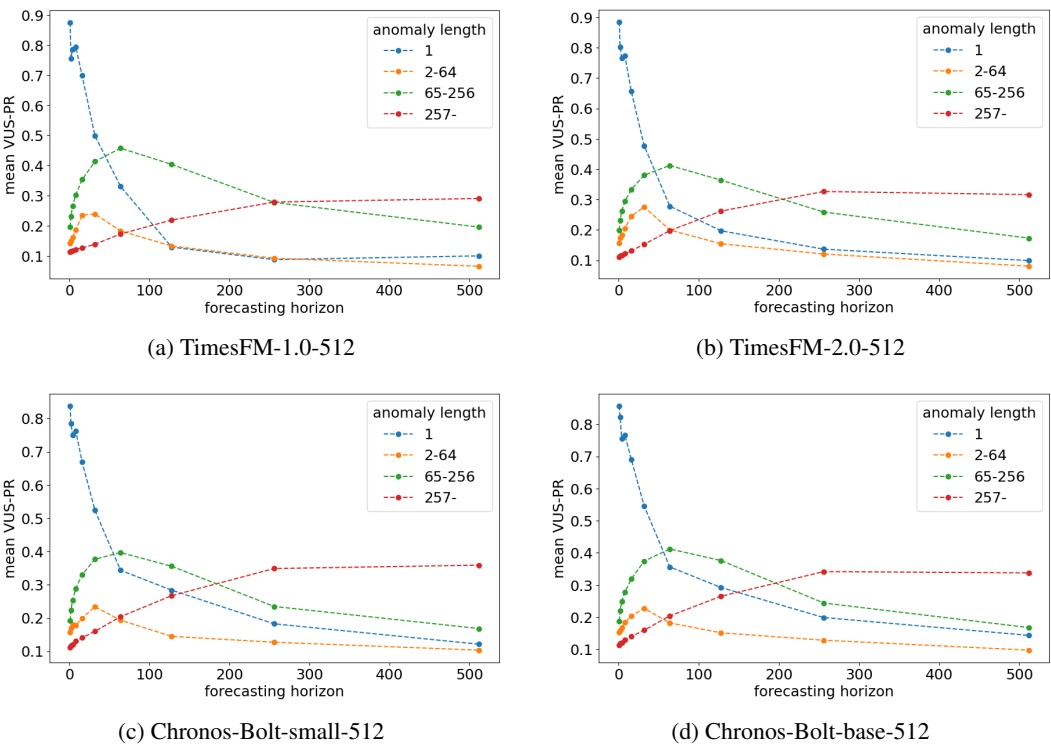

(a) TimesFM-1.0-512

(b) TimesFM-2.0-512

(c) Chronos-Bolt-small-512

(d) Chronos-Bolt-base-512

Figure 18: Mean VUS-PR scores with various forecasting horizons, evaluated on time series containing anomalies of different lengths within the TSB-AD-U evaluation set.

# M  Preliminary Results on Multivariate TSFMs

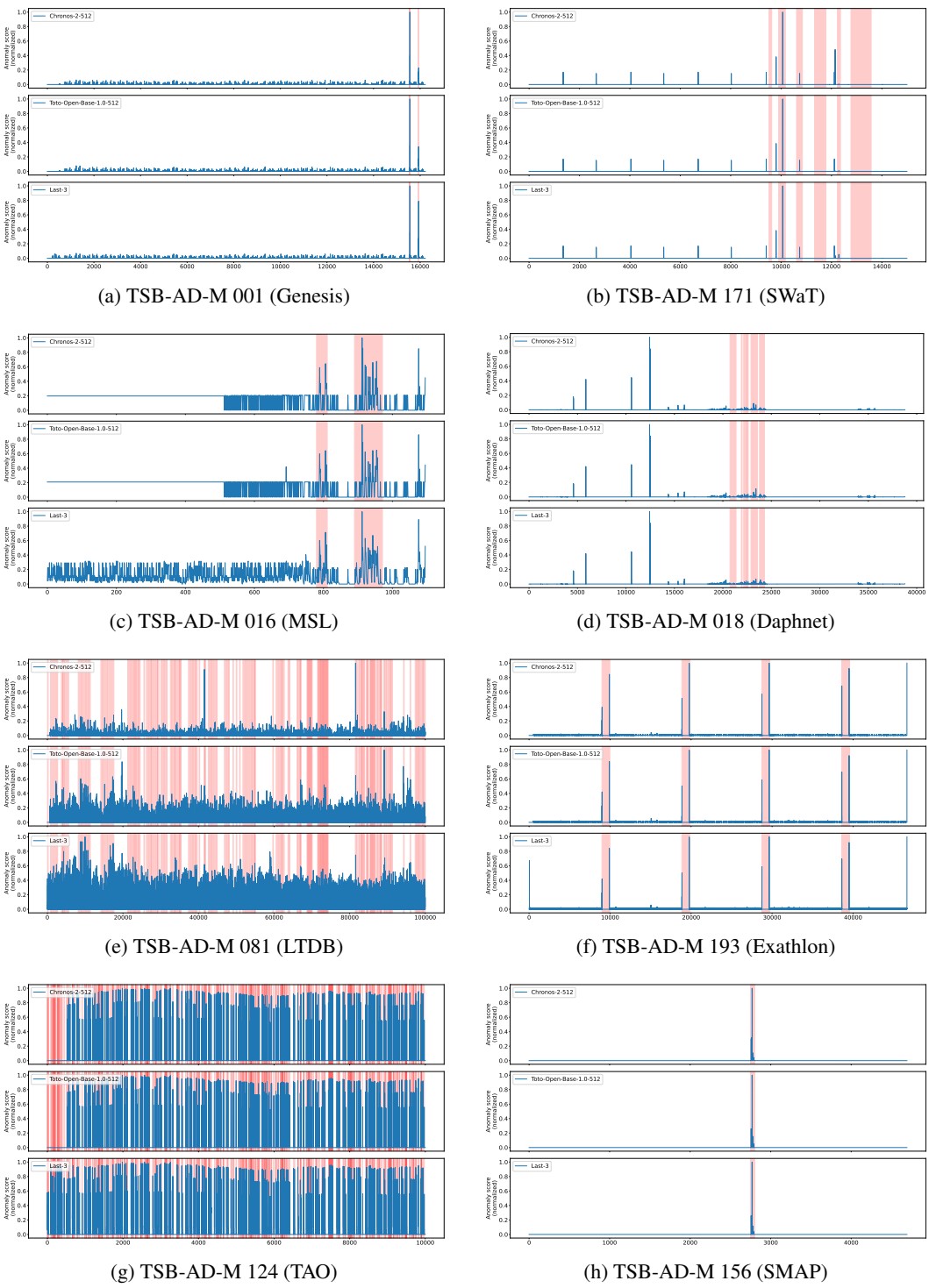

Figure 19: When employing the forecasting-based methodology for TSAD, multivariate TSFMs and the Last-3 baseline yield comparable anomaly scores. These scores were calculated as the mean squared prediction error at each time step across all channels.

