# OpenReview forum: "When Foundation Models are One-Liners: Limitations and Future Directions for Time Series Anomaly Detection"
_ICLR.cc/2026/Conference — ICLR 2026 Poster_

### Official Review · Reviewer_sn7v · 2025-10-17

**Soundness:** 2
**Presentation:** 3
**Contribution:** 2
**Rating:** 4
**Confidence:** 3

**Summary:**

The paper investigates whether TSFMs deliver reliable gains for TSAD compared with simple statistical baselines. The study evaluates reconstruction- and prediction-based TSFMs across multiple window sizes and reporting metrics, and finds limited or inconsistent advantages over one-line baselines.

**Strengths:**

The question is timely and practically important, given the rapid adoption of TSFMs and the community’s push for standard, reproducible TSAD evaluations. The experimental coverage across representative TSFMs, scoring paradigms, and window settings provides useful negative results and actionable insights.

**Weaknesses:**

- Regarding the core claim that TSFMs are not significantly better than statistical baselines, the paper relies on averaged F1/AUC-ROC without formal statistical testing.
- Regarding methodological transparency and practical guidance, it lacks of parameter counts, FLOPs, peak memory, and energy (GPU-hours), and plot accuracy versus cost to justify when TSFMs are warranted over one-line baselines.
- Regarding the explanation that prediction-based TSFMs underperform due to single-step horizons leveraging local anomaly leakage, the paper may need an ablation over the forecasting horizon H. It will be beneficial to plot anomaly-detection metrics as H increases from 1 to L/4 and L/2 (with L the window length) to verify if longer horizons better separate anomalous regimes.
- Minor comments:
    - typo “detecte anomalies” → “detect anomalies”
    - typo “requires” → “require” (line 470)

**Questions:**

Since density-based TSAD approaches [1–3] represent another important direction in anomaly detection, could density scoring offer a complementary mechanism for TSFMs?

[1] Graph-Augmented Normalizing Flows for Anomaly Detection of Multiple Time Series. ICLR 2022.

[2] Detecting Multivariate Time Series Anomalies with Zero Known Label. AAAI 2023.

[3] CaPulse: Detecting Anomalies by Tuning in to the Causal Rhythms of Time Series. 2025.

---

> ### Author Response · Authors · 2025-11-21
> **Response to Reviewer sn7v**
>
> # sn7v-W1
> > Regarding the core claim that TSFMs are not significantly better than statistical baselines, the paper relies on averaged F1/AUC-ROC without formal statistical testing.
>
> Our manuscript **does not use F1 or the AUC-ROC metric**. Instead, we use the **VUS-PR** metric, which is explicitly designed to provide a more reliable evaluation for TSAD, addressing recognized flaws in earlier metrics [1, 2].
>
> Our analysis moves **beyond aggregate metrics**: we not only presented one-to-one comparisons but also provided analysis on the underlying mechanisms for the performance similarities we found.
>
> Regarding formal statistical testing, thank you for pointing out this weakness. **Since submission we have carried out statistical analysis** to show that:
>
> 1. The performance of the TSFMs and the corresponding baselines have significantly **high linear correlation** (high Pearson correlation coefficient with extremely low corresponding p-value).
>
> 2. The effect size (Cohen’s *d*) between the VUS-PR scores of the TSFMs and the corresponding baselines is small (< 0.2), indicating **small overall performance difference**.
>
> 3. We **can exclude the possibility (at >95% confidence level) that the TSFMs outperform the corresponding baselines by more than a small margin** (0.016 for MOMENT-large, 0.0008 for MOMENT-base, and 0.008 for all forecasting-based TSFMs).
>
> **Please find the results in Appendix J in the updated manuscript. We will include an indicative part in the main text if you find it appropriate.**
>
> [1] The Elephant in the Room: Towards A Reliable Time-Series Anomaly Detection Benchmark, NeurIPS 2024.
>
> [2] VUS: effective and efficient accuracy measures for time-series anomaly detection, VLDB 2025.
>
> # sn7v-W2
> > Regarding methodological transparency and practical guidance, it lacks of parameter counts, FLOPs, peak memory, and energy (GPU-hours), and plot accuracy versus cost to justify when TSFMs are warranted over one-line baselines.
>
> We contend that a full-scale, quantitative analysis of computational costs is not strictly necessary in this context because the gap in resource requirements between TSFMs and the simple baselines is already **massive and qualitatively apparent**.
>
> For a simple quantitative comparison, running over the entire TSB-AD-U evaluation set, our baselines only use about **4 seconds** on CPU (a single core of Xeon 8360Y), while the TSFMs require **hours or even tens of hours** (depending on the model size and context window length) on a cutting-edge GPU such as A100.
>
> We list in the table below the runtime (over the entire TSB-AD-U evaluation set) of some of our experiments:
>
> | variant | GPU | approximate runtime (hours) |
> | :--- | :---: | ---: |
> | MOMENT-base-256 | P100 | 6.2 |
> | MOMENT-large-256 | V100 | 11.7 |
> | Chronos-Bolt-small-256 | V100 | 5 |
> | Chronos-Bolt-base-256 | P100 | 7.5 |
> | Time-MoE-base-256 | V100 | 17 |
> | Time-MoE-large-256 | V100 | 48 |
> | TimesFM-1.0-256 | A100 | 3.5 |
> | TimesFM-2.0-256 | A100 | 6 |
>
> # sn7v-W3
> > Regarding the explanation that prediction-based TSFMs underperform due to single-step horizons leveraging local anomaly leakage, the paper may need an ablation over the forecasting horizon H. It will be beneficial to plot anomaly-detection metrics as H increases from 1 to L/4 and L/2 (with L the window length) to verify if longer horizons better separate anomalous regimes.
>
> The case study in Table 4 of the manuscript showed that longer horizons better separate anomalous regimes **for sequence anomalies**, although a single fixed horizon is insufficient for anomalies of different lengths. This led us to propose the ensemble of short and long horizons as a future direction in Section 6.1.
>
> Beyond this case study, **we will run larger-scale experiments and get back to you with the findings.**
>
> # sn7v-W4
> > Minor comments:
> > * typo “detecte anomalies” → “detect anomalies”
> > * typo “requires” → “require” (line 470)
>
> Thank you for pointing out the typos. We have had them corrected.
>
> # sn7v-Q1
> > Since density-based TSAD approaches [1–3] represent another important direction in anomaly detection, could density scoring offer a complementary mechanism for TSFMs?
> >
> > [1] Graph-Augmented Normalizing Flows for Anomaly Detection of Multiple Time Series. ICLR 2022.
> >
> > [2] Detecting Multivariate Time Series Anomalies with Zero Known Label. AAAI 2023.
> >
> > [3] CaPulse: Detecting Anomalies by Tuning in to the Causal Rhythms of Time Series. 2025.
>
> Thank you for the suggestion. As also hinted at in [1] (Section 5.2, "the RNN can be any sequential model, such as
> the LSTM [...] and in a broad sense a transformer"), we find it a potential research idea to explore replacing the RNNs with TSFMs in the pipelines proposed by these works [1, 2], to see whether pre-trained models would provide any benefit over RNNs trained from scratch.

---

> > ### Author Response · Authors · 2025-11-28
> > **Follow-up on forecasting horizon (W3)**
> >
> > # sn7V-W3
> >
> > > Regarding the explanation that prediction-based TSFMs underperform due to single-step horizons leveraging local anomaly leakage, the paper may need an ablation over the forecasting horizon H. It will be beneficial to plot anomaly-detection metrics as H increases from 1 to L/4 and L/2 (with L the window length) to verify if longer horizons better separate anomalous regimes.
> >
> > As promised, beyond the case study in the initial manuscript (Table 4), we conducted a follow-up experiment on TimesFM-1.0-512 **with various forecasting horizons** (1, 2, 4, 8, ... 512) over the **entire** TSB-AD-U evaluation set. **The results corroborated our initial claims.** Please see **Appendix L** of the updated manuscript for the results.

---

### Official Review · Reviewer_2izu · 2025-10-31

**Soundness:** 4
**Presentation:** 4
**Contribution:** 3
**Rating:** 6
**Confidence:** 5

**Summary:**

This paper critically evaluates five families of time-series foundation models (TSFMs)—MOMENT, Chronos, TimesFM, Time-MoE, and TSPulse—for time-series anomaly detection (TSAD). The central finding is that, across model sizes and window lengths, TSFM-based anomaly scores (reconstruction/forecast errors) often perform on par with simple “one-liner” baselines (moving-window variance and squared difference), challenging the assumption that anomalies are generally harder to reconstruct or forecast. The paper then proposes practical fixes: ensembling multiple forecasting horizons, detecting anomalies in hidden representations, and modest end-to-end fine-tuning with labeled anomalies.

**Strengths:**

- Originality: Moves beyond leaderboard reporting to interrogate the mechanism behind TSFM-for-TSAD, revealing why reconstruction/forecast error can collapse to simple variance/neighbor baselines.
- Quality: Evaluates five prominent TSFM families under consistent setups (normalization, tokenization, architecture choices summarized), with direct, like-for-like baseline comparisons.
- Clarity: Effective visualizations diagnose single- vs long-horizon failures and show how representation-space detection helps.
- Significance: Practical guidance—multi-horizon ensembles, embedding-space detectors, light supervised alignment—can immediately influence TSAD deployments.

**Weaknesses:**

1. Data/setting coverage is limited. Experiments focus on univariate benchmarks; richer multivariate and domain-diverse evaluations (industrial, finance, sensors) would strengthen external validity.
2. Scope is narrow. The study centers on TSFMs vs. simple baselines; a fuller comparison against strong classical/modern TSAD methods (e.g., change-point, matrix decomposition, graph-based multivariate models) under matched conditions would be informative.
3. Statistical rigor. Many insights are qualitative; add paired tests, effect sizes, and multiple-comparison controls for key claims.

**Questions:**

1. Normalization sensitivity: If replacing z-scaling/sliding-window normalization with RevIN or sequence-level scaling, do the “TSFMs ≈ one-liners” findings materially change?
2. Horizon selection: Have you tried uncertainty-aware or autocorrelation-aware adaptive horizon selection instead of max-over-h ensembling? How does it trade off early detection vs false alarms?
3. Label granularity: If evaluated with event-level metrics (e.g., tolerant windows) instead of point-level, do core conclusions persist?

---

> ### Author Response · Authors · 2025-11-24
> **Response to Reviewer 2izu (1)**
>
> Thank you for the valuable comments. We are happy that you acknowledged our efforts to move beyond leaderboard reporting! Please find below our response.
>
> # 2izu-W1
> > Data/setting coverage is limited. Experiments focus on univariate benchmarks; richer multivariate and domain-diverse evaluations (industrial, finance, sensors) would strengthen external validity.
>
> ## Multivariate
>
> Although our experiments were on univariate benchmarks, our conclusions **can already tell how these models would perform for multivariate cases**. This is because the TSFMs analyzed in this work explicitly adopt **"Channel Independence"**. This means they **do not model cross-channel correlations**; instead, they decompose multivariate inputs into independent univariate series and process them individually. This design choice is explicitly stated by the authors of the TSFMs:
>
> * **Time-MoE**: “[… ] channel independence (Nie et al., 2023) is adopted to transform a multivariate input into univariate series.” [1]
> * **MOMENT**: “We handle multi-variate time series by independently operating on each channel along the batch dimension. Like recent studies (Zhou et al., 2023; Nie et al.,2023), we found that modeling each channel independently is an effective strategy for modeling multivariate time series.” [2]
> * **Chronos**: “In this work, we have focused on univariate forecasting of uniformly-spaced time series since it constitutes the most common of real-world time series use-cases.” [3]
>
> Because the internal mechanism of these TSFMs is strictly univariate, evaluating them on univariate data fully captures their anomaly detection capabilities. **An anomaly detection mechanism for multivariate data using these models would simply be an aggregation of univariate scores. Since we have demonstrated that TSFM performance is statistically similar to simple baselines on univariate data, it follows that aggregating these scores would just replicate the behavior of aggregating the baselines**.
>
> **The only exceptions that we know of are the latest Toto 1.0 [4] and Chronos-2 [5]**, the latter being released in October 2025, **after our manuscript was submitted**. These two latest models do try to capture inter-channel dependencies for multivariate time series. We will try to see if we can conduct some preliminary experiments to see whether the single-step horizon would also bottleneck their performance for multivariate TSAD.
>
> ## Domain-diversity
>
> Our experiments utilized the TSB-AD-U evaluation set, which is already highly heterogeneous, containing 350 time series curated from **23 diverse datasets spanning various domains** (Web Service, **Sensor**, Environment, Traffic, **Finance**, Facility, Medical, Synthetic) [6, Section B.1.1].
>
> # 2izu-W2
> > Scope is narrow. The study centers on TSFMs vs. simple baselines; a fuller comparison against strong classical/modern TSAD methods (e.g., change-point, matrix decomposition, graph-based multivariate models) under matched conditions would be informative.
>
> Previous work [6] has **already compared TSFMs against 27 classical/modern TSAD methods**. In relation to our work, we do find that the performance of the vast majority of the non-TSFM methods covered in [6] **correlate much weaker with our baselines** (see Appendix K in the updated manuscript). This is **as expected** because most of those methods do not use the pipelines to which we attribute the performance similarity with our baselines.
>
> # 2izu-W3
> > Statistical rigor. Many insights are qualitative; add paired tests, effect sizes, and multiple-comparison controls for key claims.
>
> Thank you for pointing out this weakness. **Since submission we have carried out statistical analysis** to show that:
>
> 1. The performance of the TSFMs and the corresponding baselines have significantly **high linear correlation** (high Pearson correlation coefficient with extremely low corresponding p-value).
>
> 2. The effect size (Cohen’s d) between the VUS-PR scores of the TSFMs and the corresponding baselines is small (< 0.2), indicating **small overall performance difference**.
>
> 3. We **can exclude the possibility (at >95% confidence level) that the TSFMs outperform the corresponding baselines by more than a small margin** (0.016 for MOMENT-large, 0.0008 for MOMENT-base, and 0.008 for all forecasting-based TSFMs).
>
> **Please find the results in Appendix J in the updated manuscript. We will include an indicative part in the main text if you find it appropriate.**
>
> [1] Time-MoE: Billion-Scale Time Series Foundation Models with Mixture of Experts, ICLR 2025.
>
> [2] MOMENT: A Family of Open Time-series Foundation Models, ICML 2024.
>
> [3] Chronos: Learning the language of time series, TMLR 2024.
>
> [4] This Time is Different: An Observability Perspective on Time Series Foundation Models, NeurIPS 2025.
>
> [5] Chronos-2: From Univariate to Universal Forecasting, arXiv:2510.15821.
>
> [6] The Elephant in the Room: Towards A Reliable Time-Series Anomaly Detection Benchmark, NeurIPS 2024.

---

> > ### Author Response · Authors · 2025-11-24
> > **Response to Reviewer 2izu (2)**
> >
> > # 2izu-Q1
> > > Normalization sensitivity: If replacing z-scaling/sliding-window normalization with RevIN or sequence-level scaling, do the “TSFMs ≈ one-liners” findings materially change?
> >
> > Thank you for suggesting this study. We performed it mainly with **Min-Max** and **global Z-norm**.
> >
> > We **didn’t include RevIN** for the following reason: the only difference between RevIN and window-wise Z-norm/de-norm is the introduction of a learnable affine transformation. Since the models in question (e.g., MOMENT) were not pre-trained with RevIN (at implementation level they usually just use RevIN with the affine transformation disabled), applying it in a zero-shot setting would require training these parameters from scratch. As our study focuses on the zero-shot capabilities of off-the-shelf TSFMs, introducing a training phase for normalization parameters falls outside the scope of this work.
> >
> > **We placed the results in Appendix I of our manuscript and will include an indicative part in the main text if you find it appropriate. We discuss below the results for the two types of models.**
> >
> > ## Reconstruction-based models (MOMENT)
> >
> > **The variance collapse is weakened as expected.** As shown in Equation 1 and Appendix D of our manuscript, the variance collapse relies on window-wise Z-norm and almost constant reconstruction MSE. When we switch to Global or Min-Max normalization, this **mathematical link to the local window variance is broken**, and the model deviates more from the baseline, as shown in **Figure 16 in Appendix I**.
> >
> > Note that our experiments also showed that simply changing normalization **does not improve** the overall anomaly detection performance.
> >
> > ## Forecasting-based models (TimesFM, Time-MoE, Chronos)
> >
> > As discussed in Section 5.2, we attribute the performance similarity between forecasting-based TSFMs and the squared-difference baseline to the **single-step forecasting horizon, rather than the normalization scheme**. The ablation study confirms this hypothesis: The high correlation between the model and the baseline **persists regardless of the normalization scheme**, as shown in **Figure 16 in Appendix I**.
> >
> > # 2izu-Q2
> > > Horizon selection: Have you tried uncertainty-aware or autocorrelation-aware adaptive horizon selection instead of max-over-h ensembling? How does it trade off early detection vs false alarms?
> >
> > Thank you for this valuable feedback. In Section 6.1, we opted for a max-over-h ensembling, because the maximum serves as an or-operation in which we want to flag an observation that is anomalous with respect to a short *or* long forecasting horizon. **We did not consider uncertainty-aware techniques yet, but do think this might be a promising approach and will include it in the manuscript as possible extension of our proof-of-concept**.
> >
> > # 2izu-Q3
> > > Label granularity: If evaluated with event-level metrics (e.g., tolerant windows) instead of point-level, do core conclusions persist?
> >
> > Our key findings still hold when the metric is switched from VUS-PR to Event-based F1, please see Appendix J (line 1222-1226) of the updated manuscript.

---

> > > ### Author Response · Authors · 2025-11-28
> > > **Follow-up on multivariate TSFMs (W1)**
> > >
> > > # 2izu-W1
> > > > Data/setting coverage is limited. Experiments focus on univariate benchmarks; richer multivariate and domain-diverse evaluations (industrial, finance, sensors) would strengthen external validity.
> > >
> > > As promised, we conducted preliminary case studies with Toto-1.0 and Chronos-2 using **multivariate** time series from **various** datasets. We found that, similar to the univariate case, the models' anomaly scores **remain highly similar** to those of the Last-3 baseline (when forecasting horizon = 1). Please see **Appendix M** of the updated manuscript for details.

---

### Official Review · Reviewer_nWUb · 2025-11-01

**Soundness:** 3
**Presentation:** 2
**Contribution:** 3
**Rating:** 6
**Confidence:** 4

**Summary:**

This paper critically evaluates the effectiveness of time-series foundation models (TSFMs) for anomaly detection, examining five popular model families: MOMENT, Chronos, TimesFM, Time-MoE, and TSPulse. The authors' main finding is that across different model sizes and context window lengths, these TSFMs perform no better than simple one-liner baselines—moving-window variance for reconstruction-based methods and squared-difference for forecasting-based methods. The results reveal that the fundamental assumption underlying these approaches—that anomalies are harder to reconstruct or forecast—does not hold for current TSFMs. The paper contributes a comprehensive empirical analysis demonstrating this limitation, emphasizes the importance of visualizing data and algorithm outputs rather than relying solely on aggregate metrics, and proposes three alternative directions for future research: combining multiple forecasting horizons, leveraging hidden representations for anomaly detection, and performing end-to-end fine-tuning with labeled anomalies.

**Strengths:**

1. very important question to answer
2. a lot of experiments on many TSFMs, covering both reconstructing and forecasting models
3. agree that visualization in anomaly detection is important
4. good, detailed analyses

**Weaknesses:**

1. Focus only on univariate time series. TBH, I don't agree with the argument in footnote 1 in page 2 because univariate is easier (so the baseline methods might work on univariate but not on multivariate) and TSFMs are trained on multivariate time series.
2. Need to see tabularized results to compare baselines against TSFMs to be more convincing. (seems like TSFMs are still slightly better than baselines)
3. Some of the TSFMs are not trained for anomaly detection, so comparison is not fair

**Questions:**

1. Could you clarify the conventions used in the box plots? Specifically, what do the solid and dashed lines represent—for instance, are they the mean and median?
2. On small-scale univariate time series, a specialized baseline method may perform well, but identifying the correct baseline often requires data-specific pre-analysis. Are you able to identify without pre-analysis? The value proposition of a FM lies not in outperforming a perfectly tuned baseline, but in its ability to be applied universally to any dataset and achieve robust, reasonable results without such data-specific selection.
3. It is worth testing if combining TSFMs with baseline methods improves performance. A lack of improvement would indicate that the TSFM may not be offering additive predictive power beyond the baselines

---

> ### Author Response · Authors · 2025-11-24
> **Response to Reviewer nWUb (1)**
>
> Thank you for the valuable comments. We are happy that you agree with our emphasis on visualization! Please find below our response.
>
> # nWUb-W1
> > Focus only on univariate time series. TBH, I don't agree with the argument in footnote 1 in page 2 because univariate is easier (so the baseline methods might work on univariate but not on multivariate) and TSFMs are trained on multivariate time series.
>
> While we agree that multivariate time series generally present more complex challenges than univariate ones, the TSFMs analyzed in this work explicitly adopt **"Channel Independence"**. This means they **do not model cross-channel correlations**; instead, they decompose multivariate inputs into independent univariate series and process them individually. This design choice is explicitly stated by the authors of the TSFMs:
>
> * **Time-MoE**: “[… ] channel independence (Nie et al., 2023) is adopted to transform a multivariate input into univariate series.” [1]
> * **MOMENT**: “We handle multi-variate time series by independently operating on each channel along the batch dimension. Like recent studies (Zhou et al., 2023; Nie et al.,2023), we found that modeling each channel independently is an effective strategy for modeling multivariate time series.” [2]
> * **Chronos**: “In this work, we have focused on univariate forecasting of uniformly-spaced time series since it constitutes the most common of real-world time series use-cases.” [3]
>
> Because the internal mechanism of these TSFMs is strictly univariate, evaluating them on univariate data fully captures their anomaly detection capabilities. **An anomaly detection mechanism for multivariate data using these models would simply be an aggregation of univariate scores. Since we have demonstrated that TSFM performance is statistically similar to simple baselines on univariate data, it follows that aggregating these scores would just replicate the behavior of aggregating the baselines**.
>
> **The only exceptions that we know of are the latest Toto 1.0 [4] and Chronos-2 [5]**, the latter being released in October 2025, **after our manuscript was submitted**. These two latest models do try to capture inter-channel dependencies for multivariate time series. We will try to see if we can conduct some preliminary experiments to see whether the single-step horizon would also bottleneck their performance for multivariate TSAD.
>
> # nWUb-W2
> > Need to see tabularized results to compare baselines against TSFMs to be more convincing. (seems like TSFMs are still slightly better than baselines)
>
> Regarding tabularized results, thank you for pointing out this weakness. **Since submission we have carried out statistical analysis** to show that:
>
> 1. The performance of the TSFMs and the corresponding baselines have significantly **high linear correlation** (high Pearson correlation coefficient with extremely low corresponding p-value).
>
> 2. The effect size (Cohen’s d) between the VUS-PR scores of the TSFMs and the corresponding baselines is small (< 0.2), indicating **small overall performance difference**.
>
> 3. We **can exclude the possibility (at >95% confidence level) that the TSFMs outperform the corresponding baselines by more than a small margin** (0.016 for MOMENT-large, 0.0008 for MOMENT-base, and 0.008 for all forecasting-based TSFMs).
>
> **Please find the results in Appendix J in the updated manuscript. We will include an indicative part in the main text if you find it appropriate.**
>
> # nWUb-W3
> > Some of the TSFMs are not trained for anomaly detection, so comparison is not fair
>
> We justify our comparison on two grounds. First, the core promise of **foundation** models is their ability to generalize to downstream tasks (like anomaly detection) without task-specific pre-training, e.g.:
>
> “Thus far, our exploration has centered on the problem of time series forecasting. However, several other time series analysis tasks, such as classification, clustering, and anomaly detection […] could potentially benefit from a pretrained model like Chronos.” [3]
>
> Evaluating this zero-shot capability is therefore not "unfair," but rather a test of the model's fundamental utility. Second, the baselines we employ—simple moving-window variance and squared-difference —are also not trained for anomaly detection. Comparing "untrained" TSFMs against "untrained" statistical baselines provides a level playing field.
>
> [1] Time-MoE: Billion-Scale Time Series Foundation Models with Mixture of Experts, ICLR 2025.
>
> [2] MOMENT: A Family of Open Time-series Foundation Models, ICML 2024.
>
> [3] Chronos: Learning the language of time series, TMLR 2024.
>
> [4] This Time is Different: An Observability Perspective on Time Series Foundation Models, NeurIPS 2025.
>
> [5] Chronos-2: From Univariate to Universal Forecasting, arXiv:2510.15821.

---

> > ### Author Response · Authors · 2025-11-24
> > **Response to Reviewer nWUb (2)**
> >
> > # nWUb-Q1
> > > Could you clarify the conventions used in the box plots? Specifically, what do the solid and dashed lines represent—for instance, are they the mean and median?
> >
> > Thank you for pointing this out. **The dashed lines represent the mean and the solid lines represent the median. We now mention this explicitly for Figure 5(a)  and Figure 11 in the updated manuscript.**
> >
> > # nWUb-Q2
> > > On small-scale univariate time series, a specialized baseline method may perform well, but identifying the correct baseline often requires data-specific pre-analysis. Are you able to identify without pre-analysis? The value proposition of a FM lies not in outperforming a perfectly tuned baseline, but in its ability to be applied universally to any dataset and achieve robust, reasonable results without such data-specific selection.
> >
> > 1. **The baselines are universally applied "one-liners", not tuned specialists**. We do not select baselines based on **data characteristics**. Instead, we map the baseline strictly to the **methodology** of the TSAD pipelines being evaluated, as defined in our "Experimental Setup" (Section 4).
> >
> > 2. **Our claims are not limited to specific small-scale datasets**. Our experiments utilized the TSB-AD-U evaluation set, which is highly heterogeneous, containing 350 time series curated from 23 diverse datasets spanning various domains (Web Service, Sensor, Environment, Traffic, Finance, Facility, Medical, Synthetic) [1, Section B.1.1].
> >
> > [1] The Elephant in the Room: Towards A Reliable Time-Series Anomaly Detection Benchmark, NeurIPS 2024.
> >
> > # nWUb-Q3
> > > It is worth testing if combining TSFMs with baseline methods improves performance. A lack of improvement would indicate that the TSFM may not be offering additive predictive power beyond the baselines
> >
> > We carried out this experiment for MOMENT-large and Moving-window Variance, for which the correlation between the performance of the TSFM and the baseline is the least strong (Table 6). Even with this pair, the experiments showed that ensembling the model and the baseline (max aggregation on min-max normalized scores) doesn’t lead to performance improvements over the TSB-AD-U evaluation set, as shown in the table below.
> >
> > | | mean VUS-PR (ensemble) | mean VUS-PR (baseline) | correlation | Cohen's *d* |
> > |-------|-----:|-------:|----------:|---------:|
> > | MOMENT-large-64 & Var-64 | 0.414 | 0.414 | 0.99 | -0.008 |
> > | MOMENT-large-256 & Var-256 | 0.390 | 0.386 | 0.98 | 0.059 |
> > | MOMENT-large-512 & Var-512 | 0.313 | 0.313 | 0.98 | 0.006 |
> > | MOMENT-large-1024 & Var-1024 | 0.271 | 0.273 | 0.99 | -0.036 |

---

> > > ### Author Response · Authors · 2025-11-28
> > > **Follow-up on multivariate TSFMs (W1)**
> > >
> > > # nWUb-W1
> > > > Focus only on univariate time series. TBH, I don't agree with the argument in footnote 1 in page 2 because univariate is easier (so the baseline methods might work on univariate but not on multivariate) and TSFMs are trained on multivariate time series.
> > >
> > > As promised, we conducted preliminary case studies with Toto-1.0 and Chronos-2 using **multivariate** time series from **various** datasets. We found that, similar to the univariate case, the models' anomaly scores **remain highly similar** to those of the Last-3 baseline (when forecasting horizon = 1). Please see **Appendix M** of the updated manuscript for details.

---

### Official Review · Reviewer_tjti · 2025-11-01

**Soundness:** 2
**Presentation:** 3
**Contribution:** 2
**Rating:** 4
**Confidence:** 4

**Summary:**

The paper investigates why time-series foundation models (TSFMs) underperform on time-series anomaly detection (TSAD) when used with standard error-based pipelines. The core empirical finding is that, for reconstruction-based use (e.g., MOMENT), sliding-window z-normalization makes the final anomaly score a moving-variance detector, since the normalized-space MSE is nearly constant across windows. For forecasting-based use, the paper shows single-step horizons can miss sequence-type anomalies. Across several TSFMs and baselines, simple one-line methods (moving variance, last-value/centered mean predictors) are competitive. The paper argues for (i) representation-space anomaly detection instead of error-based scoring, and (ii) more appropriate horizon settings.

**Strengths:**

- **Clear mechanistic insight**: neat derivation tying de-normalization to $(\sigma_i^2)$ and explaining the empirical collapse to variance detection.
- **Diagnosis** of a widely used pipeline (sliding z-norm + error scoring).
- **Useful negative results** that recalibrate expectations vs. simple baselines.
- **Horizon analysis** highlighting why single-step forecasting misses sequence anomalies.
- **Constructive direction** toward **representation-space** anomaly scoring.

**Weaknesses:**

- **Over-reach in generality**: conclusions for MOMENT/z-norm are at times presented as TSFM-wide without parallel derivations or tests for TimesFM/Time-MoE/Chronos/others.
- **Fairness of scope**: several forecasting models did **not** originally claim TSAD capability; the paper risks attributing pipeline failures to **model families**.
- **Framing**: occasionally reads as a critique of **TSFMs per se**, while the evidence mainly indicts the **error-based TSAD pipeline**.

**Questions:**

1. **Normalization ablations**: What happens under RevIN, MinMax, and global (non-sliding) normalization for MOMENT and at least one forecasting TSFM? Does the variance-collapse remain?
2. **Model-wise derivations**: Can you provide an analog of the de-normalization argument for TimesFM/Time-MoE (z-scaling) or explain decisively why it does/doesn’t apply?
3. **Chronos/mean-scaling**: If Chronos underperforms, is it for the **same** reason? Please dissect with controlled experiments.
4. **Representation-space TSAD**: Present systematic results (kNN/density/one-class SVM) across multiple TSFMs/datasets, not just point cases—does this consistently beat one-liners?

---

> ### Author Response · Authors · 2025-11-20
> **Response to Reviewer tjti (1)**
>
> Thank you for the valuable comments. Please find below our response to your feedback and questions. We look forward to hearing your thoughts.
>
> # tjti-W1
> > **Over-reach in generality**: conclusions for MOMENT/z-norm are at times presented as TSFM-wide without parallel derivations or tests for TimesFM/Time-MoE/Chronos/others.
>
> Our manuscript **does not** generalize the Z-normalization/variance conclusion to all TSFMs. Instead, we strictly separate our analysis and conclusions into two distinct categories based on the model's operating mechanism:
>
> 1. **Reconstruction-based models (Section 5.1)**: We explicitly **limit** the conclusion of Z-normalization causing the model to act as a variance detector **to MOMENT**. We provide a specific mathematical derivation for this phenomenon in Equation (1) and Appendix D, demonstrating how Z-normalization combined with constant reconstruction MSE leads to this behavior.
>
> 2. **Forecasting-based models (Section 5.2)**: For Chronos, TimesFM, and Time-MoE, we conducted parallel tests (see Figure 5, 11, 12) and found they function similarly to a **squared-difference baseline**, **not a variance baseline**. This squared-difference baseline amounts to naively using the mean of the neighboring observations as forecast, instead of a complex model. We explicitly attribute this failure mode to the **single-step forecasting horizon**, **not Z-normalization**. We support this empirically in Section 6.1 by showing that changing the horizon fundamentally changes the detection behavior.
>
> **We lightly revised the Introduction (line 085) to make the two conclusions further decoupled there.**
>
> # tjti-W2
> > **Fairness of scope**: several forecasting models did **not** originally claim TSAD capability; the paper risks attributing pipeline failures to **model families**.
>
> While not all model creators claimed TSAD capabilities, the research community has actively adopted this hypothesis. As noted in our Introduction, recent benchmarks [1] have claimed TSFMs "show great promise" for TSAD. Our work serves as a necessary scientific evaluation of this hypothesis. We are **not criticizing the models themselves**; rather, we are cautioning the community that **naively transferring** these models to TSAD **using standard pipelines** is ineffective.
>
> [1] The Elephant in the Room: Towards A Reliable Time-Series Anomaly Detection Benchmark, NeurIPS 2024
>
> # tjti-W3
> > **Framing**: occasionally reads as a critique of **TSFMs per se**, while the evidence mainly indicts the **error-based TSAD pipeline**.
>
> We agree that our evidence indicts the pipeline rather than the models themselves. In our manuscript, **we explicitly emphasized this distinction** in our Abstract, Introduction, Body and Conclusion.
>
> * **Abstract**: “[…] current approaches for leveraging foundation models in anomaly detection are insufficient […]” (line 025)
>
> * **Introduction**: “We show that the key assumption of reconstruction-based and forecasting-based anomaly detection – anomalies are harder to reconstruct/forecast – does not hold for TSFMs” (line 082)
>
> * **Body**: “[…] conventional reconstruction- and forecasting-based TSAD methodologies […] reduce the potential of all TSFMs to the level of simple one-liners” (line 401)
>
> * **Conclusion**: “In this work, we showed that the current approaches to apply TSFMs to TSAD are not effective” (line 477)
>
> Furthermore, **we devoted an entire section (Section 6)** to exploring how to effectively perform TSAD with TSFMs.
>
> **We lightly revised the Introduction (line 083) to make this distinction more explicit there. Please let us know if you feel there are other places being misleading.**

---

> > ### Author Response · Authors · 2025-11-20
> > **Response to Reviewer tjti (2)**
> >
> > # tjti-Q1
> > > **Normalization ablations**: What happens under RevIN, MinMax, and global (non-sliding) normalization for MOMENT and at least one forecasting TSFM? Does the variance-collapse remain?
> >
> > Thank you for suggesting this ablation study. We performed this study mainly with **Min-Max** and **global Z-norm**.
> >
> > We **didn’t include RevIN** for the following reason: the only difference between RevIN and window-wise Z-norm/de-norm is the introduction of a learnable affine transformation. Since the models in question (e.g., MOMENT) were not pre-trained with RevIN (at implementation level they usually just use RevIN with the affine transformation disabled), applying it in a zero-shot setting would require training these parameters from scratch. As our study focuses on the zero-shot capabilities of off-the-shelf TSFMs, introducing a training phase for normalization parameters falls outside the scope of this work.
> >
> > **We placed the results in Appendix I of our manuscript and will include an indicative part in the main text if you find it appropriate. We discuss below the results for the two types of models.**
> >
> > ## Reconstruction-based models (MOMENT)
> >
> > **The variance collapse is weakened as expected.** As shown in Equation 1 and Appendix D of our manuscript, the variance collapse relies on window-wise Z-norm and almost constant reconstruction MSE. When we switch to Global or Min-Max normalization, this **mathematical link to the local window variance is broken**, and the model deviates more from the baseline, as shown in **Figure 16 in Appendix I**.
> >
> > Note that our experiments also showed that simply changing normalization **does not improve** the overall anomaly detection performance.
> >
> > ## Forecasting-based models (TimesFM, Time-MoE, Chronos)
> >
> > As discussed in Section 5.2, we attribute the performance similarity between forecasting-based TSFMs and the squared-difference baseline to the **single-step forecasting horizon, rather than the normalization scheme**. The ablation study confirms this hypothesis: The high correlation between the model and the baseline **persists regardless of the normalization scheme**, as shown in **Figure 16 in Appendix I**.
> >
> > # tjti-Q2
> > > **Model-wise derivations**: Can you provide an analog of the de-normalization argument for TimesFM/Time-MoE (z-scaling) or explain decisively why it does/doesn’t apply?
> >
> > The derivation regarding z-normalization is specific to the reconstruction-based pipeline and **does not apply** to the forecasting-based pipeline. This is distinct for two reasons: first, the forecasting anomaly score is calculated **point-wise** rather than averaged over a sequence; second, the target value lies **outside** the context window rather than inside it (Figure 1). Consequently, the mathematical decomposition equating the anomaly score to 'moving-window variance' times reconstruction MSE does not hold for forecasting models.
> >
> > Instead, as discussed in Section 5.2, we attribute the performance similarity between forecasting-based TSFMs and the **squared-difference** baseline (as opposed to moving-window variance) to the limitations of the **single-step forecasting horizon, rather than the normalization scheme**. The irrelevance of the normalization method in this context is further supported by the ablation study presented in our response to Q1.
> >
> > # tjti-Q3
> > > **Chronos/mean-scaling**: If Chronos underperforms, is it for the **same** reason? Please dissect with controlled experiments.
> >
> > As demonstrated in the ablation study presented in our response to Q1 (Figure 16, Appendix I), switching the normalization method between mean-scaling and z-normalization does not alter the performance similarity between Chronos and the squared-difference baseline. Because this similarity is not due to the normalization mechanism, but the **single-step forecasting horizon**.
> >
> > **Notably**, while investigating the built-in (de)normalization modules of Chronos, we observed a discrepancy between the original Chronos architecture and the variants used in our experiments. Although the original Chronos-T5 paper [1] adopted mean-scaling, the more recent Chronos-Bolt variants (used in our main results) [have shifted to z-normalization](https://github.com/amazon-science/chronos-forecasting/discussions/264). Consequently, **the results reported in our manuscript for Chronos-Bolt already reflect performance under Z-norm. We updated Table 1 in the manuscript accordingly**.
> >
> > [1] Chronos: Learning the Language of Time Series, TMLR 2024.
> >
> > # tjti-Q4
> > > **Representation-space TSAD**: Present systematic results (kNN/density/one-class SVM) across multiple TSFMs/datasets, not just point cases—does this consistently beat one-liners?
> >
> > Our goal in section 6 is to demonstrate the **potential** of alternative methods for using TSFMs for anomaly detection. Therefore, our goal was not to provide an **extensive evaluation**. We are currently working towards an extensive evaluation of this approach **as a follow-up study**.

---

### Author Response · Authors · 2025-12-03
**Summary of Rebuttal and Revisions**

We sincerely thank all the reviewers for the insightful feedback which helps to improve this work. As suggested by the reviewers, we performed the following additional experiments/analyses and included them in the manuscript. **Their conclusion always aligns with our original findings**:

* **[nWUb-W2, 2izu-W3]** Added analysis for **statistical rigor: Table 3, Appendix J**.
* **[tjti-Q1, 2izu-Q1]** Conducted **ablation study on normalization methods: Line 307-310, Line 358-360, Appendix I**.
* **[nWUb-W1, 2izu-W1]** Conducted preliminary experiments on two **very recent multivariate TSFMs**: Toto 1.0 (NeurIPS 2025) and Chronos-2 (released in October 2025): **Line 104-107, Appendix M**.
* **[sn7v-W3]** Conducted **large-scale experiments on forecasting horizons** to consolidate observations based on individual case study in the initial manuscript: **Figure 7 and Appendix L**.
* **[2izu-W2]** Compared the baselines with **non-TSFM anomaly detectors: Section 5.4**.

We also made efforts to clarify some misunderstandings in our responses to the reviewers. Due to the earlier termination of the discussion phase, we were unable to get further feedback from the reviewers. We summarize our main arguments here (details are in individual replies and minor modifications have been applied to the manuscript to further enhance clarity).

* **[tjti-W1: Over-reach in generality]** We strictly separate our analysis and conclusions into two distinct categories based on the model's operating mechanism. We **do not generalize across the two categories**.
* **[tjti-W2 & W3: Fairness of scope & Framing]** In the initial manuscript, we explicitly emphasized that our evidence indicts the **pipeline** rather than the models themselves (**see, e.g., our research question in line 154 and our conclusion in line 531**). We also devoted **an entire section** (Section 6) to exploring how to effectively perform TSAD with TSFMs.
* **[sn7v-W1: reliance on averaged F1/AUC-ROC]** Our manuscript **does not use F1 or AUC-ROC**. Instead, we use the VUS-PR metric. **Moving beyond aggregated results is a key objective and strength** of this work (as acknowledged by Reviewer 2izu, Strength 1).

---

### Meta-Review · Area_Chair_uLDa · 2025-12-29

**Summary:**

The paper presents a systematic empirical analysis of anomaly detection on time series using time-series foundation models, revealing that standard forecasting- and reconstruction-based anomaly detection pipelines built on TSFMs often perform no better than trivial, non-learning baselines.

The paper has received generally favorable reviews. Two reviewers recommend acceptance (score 6), with one reviewer being absolutely certain (confidence 5). Two reviewers gave a rating of 4. The review of one of these reviewers (tjti) is overall positive and reads closer to a score of 6; given the strong rebuttal, it is likely that this reviewer would have increased their score. The other reviewer with a rating of 4 (sn7v) provided a relatively low-confidence review (confidence 3). This low confidence is further supported by the fact that their main critique (use of pointwise metrics for evaluation) is clearly incorrect and can be ruled out with a quick reading of the paper, raising doubts about the reliability of this review.

In my assessment, the paper provides a sound benchmark analysis and delivers valuable insights for the time-series anomaly detection community. I therefore recommend acceptance.

**Reviewer Concerns:**

Scope and framing: risk of overgeneralizing conclusions beyond specific pipelines or model settings.

Limited validation of proposed alternatives (e.g., representation-space anomaly detection), which are mostly presented as future directions.

**Reviewer Scores:**

Reviewer 2izu (score 6): likely unchanged at 6; already positive, rebuttal addresses remaining concerns.

Reviewer nWUb (score 6): likely unchanged at 6; concerns addressed, overall stance already favorable.

Reviewer tjti (score 4): likely increased to 6 after clarifications on scope, normalization, statistics, and horizon ablations.

Reviewer sn7v (score 4): possibly increased to 6; several core critiques addressed, but some reservations may remain.

---

### Decision · Program_Chairs · 2026-01-26

Accept (Poster)